# Spatial biology using single-cell mass spectrometry imaging and integrated microscopy

Alexander Potthoff [1], Jan Schwenzfeier [1], Marcel Niehaus [2], Sebastian Bessler [1], Emily Hoffmann [3], Oliver Soehnlein [4], Jens Höhndorf[2], Klaus Dreisewerd [1] & Jens Soltwisch [1] ✉

Spatial biology pursues the analysis of cells in their native microenvironment, often with regard to morphology and gene- or protein expression. The spatial analysis of lipid and metabolic profiles by matrix-assisted laser desorption/ ionization mass spectrometry imaging (MALDI-MSI) can add another layer of molecular information. A seamless integration of MSI data with other (multi-modal) methods at a single-cell level is often hampered by insufficient co-registration and resolution. Here, we introduce a MALDI-MSI based method that integrates in-source bright-field and fluorescence microscopy, allowing for a coupled (sub-)cellular investigation of the same sample in both mod-alities. Presented examples from cell culture and tissue analysis include the visualization of intracellular lipid distributions in macrophages during pha-gocytosis, and the heterogeneity of lipid profiles of tumor infiltrating neu-trophils correlated to their individual microenvironments. The achieved combination of lipid profiling with morphological features and protein expression on the single-cell level constitutes a powerful method for cell biology.

Two cells in a tissue or a cell culture are never exactly identical, even if they share the same genetic heritage and microenvironment. This heterogeneity can be a consequence of differences in age, ontogeny, or tissue imprinting, resulting in alterations of morphology, gene expression, molecular composition, and ultimately function. In living systems, individual cells are part of interdependent multicellular structures, i.e., tissues, and undergo a constant exchange within local regulatory networks as part of the almost inestimable complexity that is life. Thus, to characterize the identity and state of a cell, the description should follow a holistic approach and include information from a wide range of analytical techniques[1].

In recent years, single-cell RNAseq and spectral flow cytometry have emerged as valuable tools to profile cellular heterogeneity. Similarly, advances in single-cell proteomics provide information on the heterogeneity on the protein level, albeit with much lower throughput[2,3]. Under the umbrella of spatial biology, the emergence of spatial-omics technologies including deep visual proteomics, multi-plexed immunohistochemistry and spatial RNAseq have added infor-mation on morphology and the spatial relationship of cells[4–7].

A complementary addition to these techniques is the analysis by high spatial resolution mass spectrometry imaging (MSI). Importantly, this method can augment information on active transcripts and pro-tein expression with the resulting downstream production of lipids and further metabolites, as well as glycans and peptides[8]. The most widely used MSI approach is based on matrix-assisted laser desorption ionization (MALDI). Herein, finely focused laser pulses are applied to pre-defined pixel positions on the sample surface, resulting in spatially defined mass spectra that can be processed to reveal signal intensity maps for up to hundreds of analytes differentiated based on their molecular weight. To increase signal intensity for a wide range of

[1]Institute of Hygiene, University of Münster, Münster, Germany. [2]Bruker Daltonics GmbH & Co. KG, Bremen, Germany. [3]Clinic of Radiology, University of Münster, Münster, Germany. [4]Institute of Experimental Pathology, University of Münster, Münster, Germany. ✉e-mail: jenssol@uni-muenster.de

analyte classes, the technique can be amended with laser postionization (MALDI-2)[9].

Typically, eukaryotic cells stretch over distances between 10 and 100 μm and display a wide variety in shape and form. Prominent sub-cellular structures like the nucleus or other large organelles, but also morphological features, show sizes between 3 and 10 μm. Their spatially resolved analysis, therefore, requires pixel sizes of no more than $1 \times 1 \mu m^2$. The current bench-mark figure on the pixel size of MALDI-MSI, however, is at about $5 \times 5 \mu m^2$. Less established experimental techniques dedicated to high resolving power, such as transmission-mode or inverse geometry (t-)MALDI-2-MSI and others, enables pixel sizes $\leq 1 \times 1 \mu m^2$ and can thus fulfill this criterion[10–16].

Recent applications of MALDI-2 and t-MALDI-2-MSI for single-cell mass spectrometry have revealed a strong molecular heterogeneity for lipids and metabolites within clonal populations of cells and seemingly homogeneous tissue regions, demonstrating its high potential for cell biology research[17–20]. To contextualize these metabolic phenotypes with existing knowledge on the single-cell level, however, MALDI-MSI has to be connected to prevalent microscopy techniques conducted on the same slide. To date, the general application of this contextualization on the same slide is strongly limited by three main factors: The spatial resolving power of MALDI-MSI, precise co-registration with optical modalities, and automation of data analysis. Especially in the context of tightly packed cellular systems, such as tissue, shortcomings of the first two factors lead to erroneous or ambiguous assignment of mass spectrometric information to individual cells. To date, this restricts the analysis on the single cell level directly from tissue to laborious manual annotation of comparably low numbers of cells, while automated analysis is limited to cellular neighborhoods rather than single cells[17,18,21].

To address these challenges, we here present the development of hard and software solutions on a prototype ion source coupled to an orthogonal time-of-flight (oTOF) mass analyzer. Our approach enables the generation of unambiguous cell-specific molecular and morphometric information of statistically relevant numbers of cells directly from tissue. It utilizes transmission mode for ultra-high spatial resolution and is equipped with laser postionization (MALDI-2) to increase sensitivity for the analysis of lipids and metabolites[9,15,22,23]. While previous implementations of t-MALDI ion sources have only hinted at the potential to utilize the employed optical components for microscopy of the sample inside the ion source[15,24], we here make full use of this option by including in-source bright field (BF) as well as fluorescence microscopy (FM).

## Results
### Description and benchmarking of transmission MALDI-2-MSI with integrated scanning microscopy

A full technical description of the optical setup and preparation protocols is presented in the methods. In short, the ion source of a Bruker timsTOF-fleX MALDI-2 was amended for t-MALDI analysis. Dedicated lighting strategies enable the use of BF as well as FM inside the ion source (Fig. 1a,b). Importantly, laser delivery and in-source microscopy share essential components of the optical beam path and for stage movement. In the xy-plane (sample surface), the site of laser ablation is therefore directly linked to a specific position on the optical microscope image. Different from other instrumentation that combines MALDI-MSI with microscopy in one instrument[25,26], this design feature inherently co-registers both modalities by utilizing the same coordinate system for in-source microscopy and MALDI-MSI (Fig. 1a). Next to providing highly resolved orientation to set up imaging runs, FM-images acquired inside the ion source constitute a unique and essential link for precise co-registration to other optical modalities. In the z-direction, the inherent coupling of the focal plane of optical microscopy and laser delivery enables a fast and easy setup of experiments based on autofocusing routines.

To enable FM analysis prior to MSI analysis, we have developed dedicated staining protocols optimized to omit or strongly reduce chemical alterations and spatial delocalization of the analyte content of interest. In parts these are based on protocols developed for the analysis of cell culture[17,20]. To control for the adverse effect of immunofluorescence (IF) staining on overall performance reported in the literature[27], the specifically developed methods and pipelines were benchmarked with regard to microscopic image quality, conservation of chemical and morphological integrity, as well as mass spectrometric depth of information.

For this, we conducted combined FM and t-MALDI-2-MSI on coronal cryo-sections of fresh-frozen mouse cerebellum. This tissue exhibits histological features across different lateral scales, each presenting distinct lipid profiles. It has been intensely used as a model system across a wide range of microscopic and MSI methods[28–31] and is well characterized with regard to its lipidome based on the analysis of lipid extracts[32]. Details on sample preparation and data acquisition are described in the methods.

To inspect and demonstrate the quality of our dedicated staining-based FM analysis on the same tissue section prior to t-MALDI-MSI analysis, we employed small molecule stains, targeting the nuclei and F-actin, as well as IF to target calbindin expressed in Purkinje cell bodies (Fig. 1c). Slide scanning FM recorded prior to matrix application in an external microscope (VS200, Evident) at 50X displays the expected structures and morphology for all three staining methods in high quality and corroborates the feasibility of the employed protocols (Fig. 2a–c).

After matrix application by resublimation, the sample was transferred into the MALDI ion source, followed by in-source slide scanning FM (50X, DAPI channel). Utilizing the FM image of the DAPI channel generated inside the ion source as an intermediate enables a straightforward co-registration of external FM images with MALDI-MSI (Supplementary Fig. 1a). In this, our approach omits the use of fiducial markers across modalities, relieving an inherent limitation of conventional methods. Difficulties in the identification of these "cross-modality" markers that need to represent the exact same spatial location in conventional methods have been described to limit the precision and spatial fidelity of co-registration of FM and MSI on a cellular or sub-cellular level, especially for large areas of interest[21,33,34].

Subsequently, t-MALDI-2-MSI data at a pixel size of $1 \times 1 \mu m^2$ were acquired in positive ion mode. Equivalent data in the negative ion mode was recorded on a neighboring section. Facilitated by the choice of matrix and the use of MALDI-2, mass spectra in the positive mode produce more depth of information, and this ion mode was used for all further experiments (Supplementary Table 1). Overview and zoomed in regions of the resulting MS images display distinct intensity distributions demonstrated on the example of ion signals of two different phosphatidylcholines (PC) ([PC(38:6) + H]⁺ at $m/z$ 806.57 and [PC(40:6) + H]⁺ at $m/z$ 834.60) and the molecular ion signal of the Hoechst stain of the nuclei at $m/z$ 452.23 (Fig. 2c).

Direct comparison of t-MALDI-2-MSI with microscopy confirms that morphological features at the different lateral scales are largely retained during the matrix application step and subsequent MSI analysis (Fig. 2b). The overlay of both modalities reveals a strong correlation with histological features of the white matter and granular layer, discernible based on the F-actin staining and the nuclei. Similarly, signal intensity distribution of PC(40:6), associated with the Purkinje cell layer[35] shows good correlation with the IF-based specific staining of these structures (Fig. 2b). Overall, deviations for co-registration were found to be less than 1 μm (Supplementary Fig. 1b, c and Supplementary Note 1), corroborating the precision and fidelity of our approach.

To explore the depth of information for lipid analysis available from pre-stained t-MALDI-2-MSI, mass spectral information was matched against data from an unstained tissue section of the same brain.

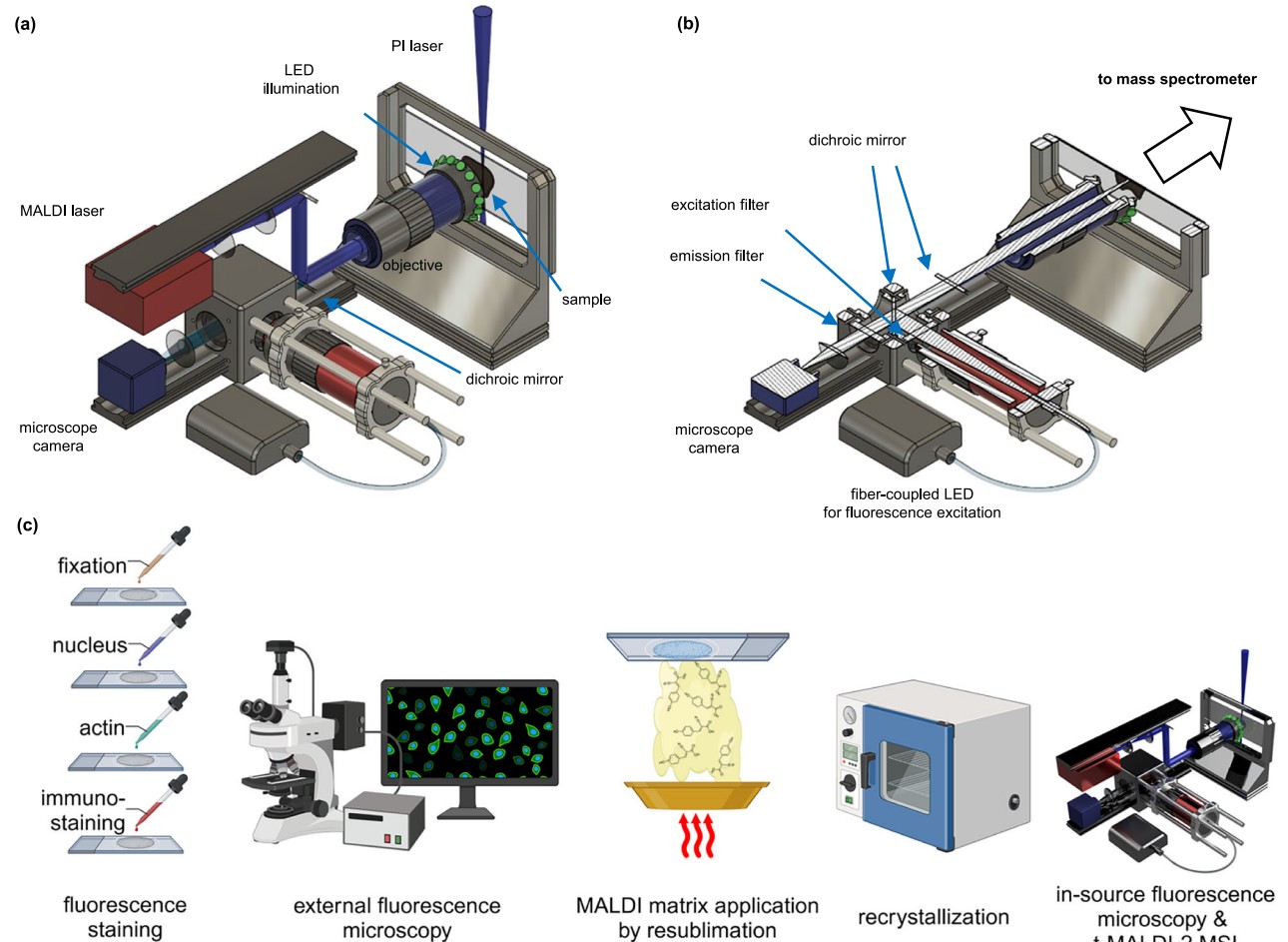

**Fig. 1 | Transmission MALDI-2-MSI with integrated scanning microscopy for subcellular resolving power. a** A schematic drawing of the t-MALDI-2 setup illustrates the general layout. The sample is illuminated homogeneously by an LED ring. Light is collected by the microscope objective and focused onto the sensor of a digital camera. The MALDI laser-beam is co-aligned using a dichroic mirror and focused onto the sample by the same objective. Ions generated by the MALDI process with optional MALDI-2 are extracted to a mass analyzer. **b** A horizontal cut through the schematic displays the fluorescence microscope components. Fluorescence is excited by a fiber-coupled LED and filtered by an excitation filter, a

dichroic mirror and an emission filter. ((**a**) and (**b**) are renderings created by the authors in Autodesk Fusion 360.) **c** To enable fluorescence microscopy and t-MALDI-2-MSI on the same sample, different staining steps are performed following a brief fixation. Samples are imaged with an external slide scanning fluorescence microscope, and are then prepared for MALDI-MSI by homogeneous coating with MALDI matrix (Supplementary Fig. 5 A, B, D). To enable high fidelity co-registration of the fluorescence microscopy image, a second fluorescence image is recorded inside the ion source prior to t-MALDI-2-MSI (Supplementary Fig. 1a, d-e).

Overall, the staining protocol does not introduce a decrease in depth of lipid information with regard to the tentatively assigned lipids species (Supplementary Table 2, Supplementary Fig. 2). While alkali metal cations of lipids dominate the spectra of the naïve tissue sections staining, washing steps that are part of the staining protocols shift the ionization towards protonation (Supplementary Fig. 2). Interestingly, only the naïve tissue produces sizeable signal intensities of phosphatidic acids. Their appearance has been associated with in-source fragmentation of alkali metal cations of other phospholipids[36].

In addition, lipid information from t-MALDI-2-MSI was compared to a previously published full lipidomics profile for murine cerebellum derived from bulk[32]. This analysis reveals that ~82% of the reported molecular lipid content of cerebellum is detected in the t-MALDI-2 measurement (Supplementary Note 2). This amounts to 66 annotated *ion species* that produce contrast-rich intensity distributions in t-MALDI-2-MSI in positive and/or negative ion mode measurements (Supplementary Table 1). Comparison of the detected signal intensities with the molar content of the different lipoforms of each of the detected lipid classes shows a good agreement with the literature[32] (Supplementary Fig. 3 and Supplementary Note 2). Notably, this includes lipid classes, such as phosphatidylethanolamine (PE) and

phosphatidylserine (PS) that have been described to be affected if more extensive chemical fixation is used during sample preparation[37], corroborating the aptitude of our specifically developed protocols.

## Lipid analysis at sub-cellular resolving power

To demonstrate the potential of histology guided t-MALDI-2-MSI for the analysis of sub-cellular structures, we investigated THP-1-derived macrophages. Cells were cultured and differentiated directly in chamber slides. Freshly differentiated cells were incubated with pHrodo red *E. coli* bioparticles for 2 h. Upon phagocytosis, these particles become fluorescent and pinpoint acidified phagolysosomes inside the cells. For analysis, cells were briefly fixed and their actin filament and nuclei stained following MALDI compatible protocols previously reported[17].

Slide scanning FM on an external slide scanner prior to coating with MALDI matrix reveals heterogeneously pronounced phagocytosis with some cells showing a large number of internalized bioparticles (Fig. 3a).

After coating with MALDI matrix using resublimation, in-source FM (50X) and t-MALDI-2-MSI data at $1.5 \times 1.5 \, \mu m^2$ pixel size in positive ion mode were collected for a few hundred cells (Fig. 3b). Image co-

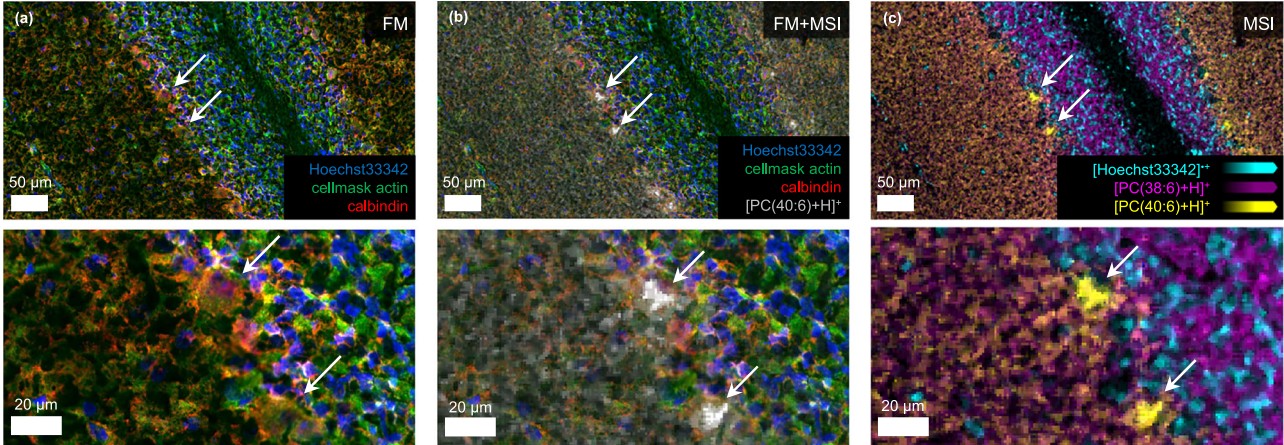

**Fig. 2 | External FM of mouse cerebellum with co-registered t-MALDI-2-MSI signals. a** An external fluorescence image of an 8 μm thick section of a mouse cerebellum recorded prior to matrix application using Hoechst 33342 (nuclei, DAPI channel, blue), CellMask Green Actin Tracking Stain (actin, FITC channel, green) and Alexa Fluor 594 Anti-calbindin antibody (calbindin, TxRed channel, red). The different layers of the brain (molecular layer, granular layer, and arbor vitae), as well as the Purkinje cells are well distinguishable. **b** The external FM image is overlaid with the ion signal intensity distribution of [PC(40:6) + H]⁺ at *m/z* 834.60 (gray) revealing a successful co-registration of both modalities. **c** The overlay of three ion signal intensities of the molecular ion of Hoechst 33342 at *m/z* 452.23 (cyan), [PC(38:6) + H]⁺ at *m/z* 806.57 (magenta) and [PC(40:6) + H]⁺ at *m/z* 834.60 (yellow), affirms the ability to produce images of high contrast while preserving and reproducing the biological structure of the sample.

registration between the externally generated FM image and MSI data was performed by using the in-source FM data as an intermediate (Supplementary Fig. 1d). The overlay of both modalities corroborates a high-fidelity co-registration on the level of the employed MALDI pixel size. Areas visualized by the pHrodo red stain in FM accurately co-align with the distribution of specific lipid signal intensities in MSI, revealing little to no leakage or diffusion of lipid content or other preparation artifacts that may compromise (sub-)cellular integrity (Fig. 3c,d)[22].

To investigate the molecular distributions specifically related to phagolysosomes, we used subcellular spatial information provided by the fluorescence signal produced by the uptake of pHrodo bio-particles. Utilizing the high-fidelity co-registration, we overlayed this histological information to t-MALDI-2-MSI data and identified those pixels that correlate with phagolysosomes to generate phagolysosome-specific mass spectra. (Fig. 3c,d). Lipid annotation was based on previously published results for THP-1 derived macrophages and *E. coli*, respectively[20,38].

Comparative analysis of the resulting phagolysosome-specific molecular information revealed an increase of signal intensity for numerous lipid species (Fig. 4). Confirmed by the respective MS images, lipids, such as PE(33:1) and PE(34:2) spatially correlate with the phagolysosomes identified by fluorescence (Fig. 3c,d). Both PE species are abundantly expressed in the membrane of *E. coli* cells[38]. Notably, PE(33:1) is not detected in non-phagocytosing THP-1 cells at all (Fig. 3b–d, red trace). Its presence, confirmed by additional on-cell MS/MS experiments (described in Methods), provides a direct link to the phagocytized pathogen material and corroborates the highly accurate co-registration of both modalities.

Other tentatively assigned lipid species linked to the phagolysosomes can be attributed to degradation of *E. coli*-derived phospholipids during phagocytosis[39,40]. These degradation products are most likely the result of enzymatic activity and reactions with reactive oxygen species inside the phagolysosomes. They include lipid metabolites, such as the diacylglycerol DG(33:1), which can be directly linked to the enzymatic cleavage of PE(33:1) and lipid species tentatively identified as lysophosphatidylglycerols (LPG). The latter can be linked to the enzymatic breakdown of phosphatidylglycerols (PG) and cardiolipins (CL) both highly expressed in the membrane of *E.coli*[38,40] (Fig. 4).

## Automated single-cell analysis of lipids in their histological context in tumor tissue

In a second exemplary study, we applied our methods to the single-cell molecular analysis of tumor tissue. This tissue type typically contains a large number of different cell types in a wide range of states, translating to a wide variety of metabolic microenvironments (tumor microenvironment (TME)) of different size and shape and a highly heterogeneous lipid profile across the tissue[41].

The employed 4T1 mouse mammary carcinoma model is highly malignant with locally aggressive growth and exhibits metastasis to regional lymph nodes and distant organs, particularly lung, liver, and bone[42,43]. It has diverse inflammatory TMEs with a large number of infiltrating immune cells, such as T-cells, macrophages, neutrophils, and natural killer cells present[44,45]. The site and cell-specific molecular profile of these immune cells is of great importance for their fate and function and is believed to be in constant interdependence with their microenvironment[46]. Consequently, the investigation of this crosstalk not only requires a molecular analysis at the cellular level but also an assessment of the immediate vicinity.

For analysis, a heterogeneous area of interest that includes a broad range of TMEs was selected based on H&E staining and histological annotation (Fig. 5a). A consecutive section was prepared using MALDI-compatible staining that targets cell nuclei, F-actin, and three IF stains that selectively target immune cells, (CD45), neutrophils (Ly6G), and T3 neutrophils (DcTRAIL-R1)[46,47]. All channels were recorded on the complete section using external slide scanning FM at 50X before matrix application (Fig. 5b). Co-registration with t-MALDI-2-MSI was based on the DAPI channel recorded inside the ion source after matrix application as an intermediate (Supplementary Fig. 1e). After microscopy, t-MALDI-2-MSI data was recorded in positive ion mode in seven separate measurements across the region of interest at a pixel size of 1 × 1 μm² resulting in a total of approx. 7 Mpixels. For data analysis, we selected 63 molecular ion species annotated to individual lipid species based on exact mass ( < 3 ppm) (see Supplementary Fig. 4 for selected images and Supplementary Table 3).

Spatial distribution for a number of glycero- and glycerophospholipids such as [PC(34:1) + H]⁺ at *m/z* 760.58 reveal a sizeable heterogeneity across the tumor sample (Fig. 5c, Supplementary Fig. 4). Signal intensity of the ether lipid [PC(O-32:0) + H]⁺, detected at *m/z*

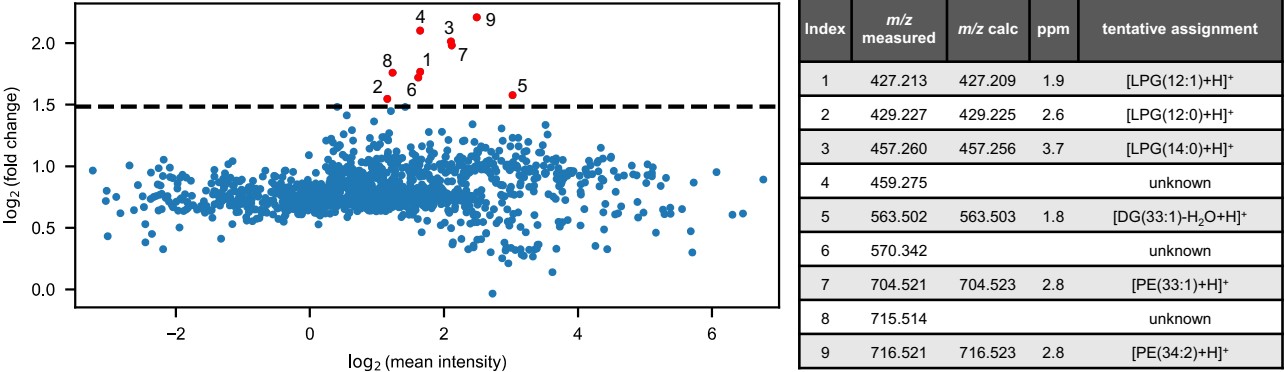

**Fig. 3 | Lipid analysis of cultured cells at sub-cellular resolving power.**
**a** Fluorescence image of THP-1 derived macrophages recorded before MALDI matrix application with a slide scanning microscope using Hoechst 33342 (nuclei, DAPI channel, blue), CellMask Green Actin Tracking Stain (actin, FITC channel, green) and pHrodo (phagolysosomes, Cy3 channel, red). **b** An overlay of three different ion signal intensity distributions measured using t-MALDI-2-MSI with $1.5 \times 1.5 \, \mu m^2$ pixel size in positive ion mode of the molecular ion of Hoechst 33342 ($m/z$ 452.23, blue), [PE(33:1) + H]$^+$ ($m/z$ 704.52, red) and [PC(34:1) + H]$^+$ ($m/z$ 760.58,

green) reveals a strong cell-to-cell heterogeneity. **c** and **d** Zoom in from (**a**) and (**b**), marked in each as a blue and red square, respectively, demonstrates intracellular differences in lipid distribution. To investigate the intracellular molecular distributions related to phagocytosis, masks for the regions with pHrodo$^+$ (red) and remaining cell (green) are created based on the FM image. The outlines of the masks identified by the pHrodo stain are overlayed with the MSI results to demonstrate the quality of co-registration.

| Index | $m/z$ measured | $m/z$ calc | ppm | tentative assignment |
|-------|----------------|------------|-----|----------------------|
| 1 | 427.213 | 427.209 | 1.9 | [LPG(12:1)+H]$^+$ |
| 2 | 429.227 | 429.225 | 2.6 | [LPG(12:0)+H]$^+$ |
| 3 | 457.260 | 457.256 | 3.7 | [LPG(14:0)+H]$^+$ |
| 4 | 459.275 | | | unknown |
| 5 | 563.502 | 563.503 | 1.8 | [DG(33:1)-H$_2$O+H]$^+$ |
| 6 | 570.342 | | | unknown |
| 7 | 704.521 | 704.523 | 2.8 | [PE(33:1)+H]$^+$ |
| 8 | 715.514 | | | unknown |
| 9 | 716.521 | 716.523 | 2.8 | [PE(34:2)+H]$^+$ |

**Fig. 4 | Comparative region-specific mass spectral data of phagolysosomes.**
Comparative analysis of the resulting region-specific mass spectral data from masks as shown in Fig. 3c,d reveal an increase of signal intensity for a number of lipid

species inside the pHrodo$^+$ regions. Signals with a log$_2$ fold change above 2.8 are marked in red. Identified $m/z$-values are denoted in a table with tentative assignments, where applicable.

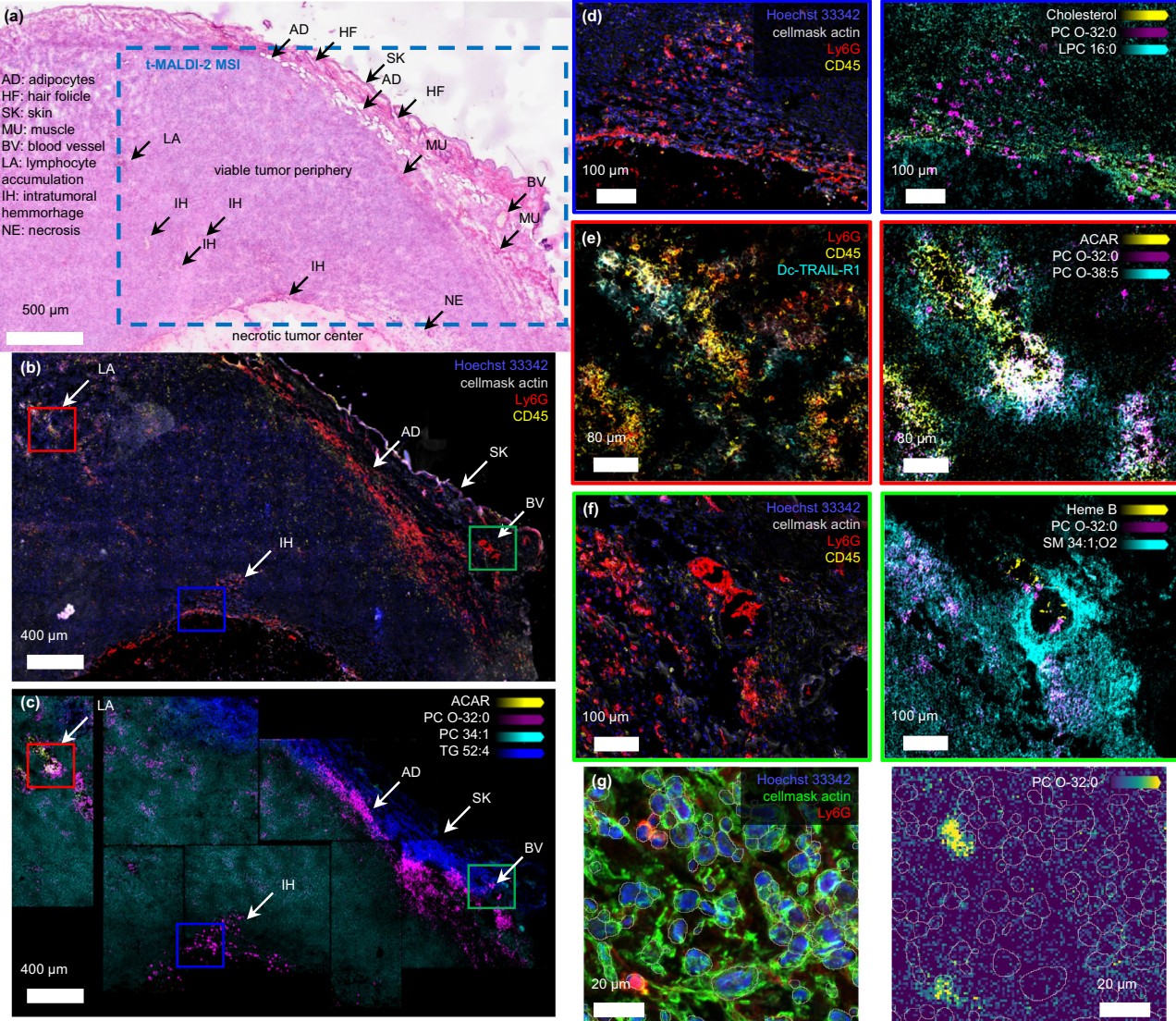

**Fig. 5 | Automated single-cell analysis of lipids in their histological tissue context. a** Annotation of an H&E stain of a 4T1 tumor section; ROI of the t-MALDI-2-MSI conducted on a consecutive section marked in blue. **b** FM using Hoechst 33342 (nuclei, DAPI channel, blue), CellMask Green Actin Tracking Stain (actin, FITC channel, gray), IF stains for neutrophil cells (Ly6G, TxRed channel, red), immune cells (CD45, Cy5 channel, yellow), and T3 neutrophils (DcTRAIL-R1, Cy7 channel, not pictured in (**b**)) prior to MALDI matrix application. **c** t-MALDI-2-MSI overlays in positive ion mode of acylcarnitine [ACAR 18:0 + H]⁺ (m/z 428.38, yellow), [PC(O 32:0) + H]⁺ (m/z 720.59, magenta), [PC(34:1) + H]⁺ (m/z 760.58, cyan) and [TG 52:4 + H]⁺ (m/z 855.75, blue) corroborating histological annotations from (**a**), (**b**). **d** Zoom-in of blue squares (**b**) and (**c**). FM shows Hoechst (blue), CellMask (gray), neutrophils (red) and immune cells (yellow). MSI overlays: [cholesterol-H₂O + H]⁺ (m/z 369.35, yellow), [PC(O-32:0) + H]⁺ (m/z 720.59, magenta), [LPC(16:0) + H]⁺ (m/z 496.34, cyan). **e** Zoom-in of red squares in (**b**) and (**c**). FM shows neutrophils (red), immune cells (yellow) and T3 neutrophils (cyan). MSI overlays: [ACAR 18:0 + H]⁺ (m/z 428.38, yellow), [PC(O-32:0) + H]⁺ (m/z 720.59, magenta) and [PC(O-38:5) + H]⁺ (m/z 794.61, cyan). **f** Zoom-in of green squares in (**b**) and (**c**). FM shows Hoechst (blue), CellMask (gray), neutrophils (red) and immune cells (yellow). MSI overlays: Molecular ion of Heme B (m/z 616.18, yellow), [PC(O-32:0) + H]⁺ (m/z 720.59, magenta) and [SM(34:1;O2) + H]⁺ (m/z 703.57, cyan). **g** 63,408 cell masks based on the FM image inside the complete ROI. Zooms show cell masks overlaid with FM and MSI. For MSI ion signal intensity of [PC(O 32:0) + H]⁺ at m/z 720.59 selected exemplarily.

720.59, correlates with the neutrophil-specific Ly6G-stain (Fig. 5d–f). This is consistent with a significant overexpression of this lipid species in mouse neutrophils reported previously[48,49].

Other specific signal intensity distributions also directly correlate with structures visible in the microscopy data. As expected[50], signal intensity for triacylglycerides (TG) such as [TG(52:5) + H]⁺ correlates with the adipose tissue (Fig. 5c). Acylcarnitine, detected at m/z 428.38 as [ACAR(18:0) + H]⁺ is an important intermediate in lipid metabolism and a reported marker for hypoxia[51]. Its signal intensity distribution correlates with the detection of tumor-associated T3 neutrophils identified by the DcTRAIL-R1 staining that have been described to be found in hypoxic areas[46,47] (Fig. 5e). Heme B, detected at m/z 616.18, is

directly associated with erythrocytes and, hence, serves as marker for hemorrhage or blood vessels[52] (Fig. 5f).

While general assignment of lipid profiles to TMEs has been reported previously, the approach described here allows for the analysis of single cells within their tissue environment in statistically relevant numbers. For this we adapted our previously reported workflow for the analysis of single cells in culture to the analysis of tissue sections[17,20]. Robust and reliable cell segmentation, necessary for automated analysis[21], was provided based on the external FM images using Mesmer by DeepCell (Fig. 5g)[53]. This resulted in segmentation masks for each cell and enabled the calculation of a variety of morphometric parameters using CellProfiler[54].

To produce single-cell mass spectra, cell segmentation masks were projected on the t-MALDI-2-MSI data, again taking advantage of the inherent link between in-source FM and the acquired MSI data (Supplementary Fig. 1e). Of note, already small inaccuracies of the co-registration on the order of a few micrometer as well as increased pixel size can lead to larger ambiguities in the assignment of MS data to specific single cells[17,21]. An overlay of the projected masks with the lipid neutrophil marker $[PC(O-32:0) + H]^+$ and comparison to the specific Ly6G staining demonstrates the high-fidelity co-registration and corroborates the ability of our combined method to deduce single-cell mass spectra (Fig. 5g).

Single-cell molecular data is generated for all 63 annotated lipid species by summing the respective signal intensities over all pixels allocated to a specific cell using the respective mask. Subsequently, all values are normalized to the sum of all signal intensities for the respective cell.

Cell segmentation and subsequent data analysis resulted in combined morphometric and molecular data for 63,408 individual cells. Notably, this analysis retains the histological context of all cells and includes parameters describing location within the tissue and fluorescence intensity for all employed channels. This enables the identification of CD45$^+$, Ly6G$^+$, and DcTRAIL-R1$^+$ cells based on intensity thresholding for each specific channel. For visualization, this information can be projected onto the segmentation canvas, thereby identifying the location and local density of different sub-groups of immune cells in the tissue section (Fig. 6a).

Processing of the combined IF and MSI single-cell data enabled spatial biology analysis on the level of lipids and metabolites, by classifying cells based on molecular similarity. In this context, analysis on the single-cell basis is especially important for scarce or isolated cells like specific immune cells because their molecular profile may substantially differ from the surrounding TME. This type of analysis is demonstrated on the example of infiltrating neutrophils inside the 4T1 tumor section.

To classify neutrophils based on their molecular profile independent of their microenvironment and vice versa, single-cell data were split into two groups based on Ly6G staining. The resulting subgroups contained 3410 Ly6G$^+$ neutrophils and 59,998 Ly6G$^-$ cells, respectively. Both groups were independently analyzed using Uniform Manifold Approximation and Projection (UMAP) and appropriate clustering algorithms.

To identify molecularly similar cell types within the tissue, all Ly6G$^-$ cells were subjected to k-means clustering, resulting in eight discernible clusters. By assigning a specific color to each cluster, the outlines of each cell can be filled in with the respective color and projected back onto their original location (Fig. 6b).

The resulting map of classified cells displays clearly discernible tissue micro-environments where molecularly similar cells cluster in specific areas of the tissue. Clusters IV, V, and VIII contain cells of the skin, the adipose tissue, and the wider tumor periphery, respectively. Inside the viable tumor tissue, a clear separation into TMEs reveals a layer-like structure with frayed edges (clusters II, III, and VI). Cluster VII contains cells directly adjacent to the tumor edges and away from the adipose tissue, but also cells inside the tumor close to its necrotic core. Cluster I contains cells allocated to hypoxic regions of the tumor.

The direct connection to the morphometric data allows to characterize each cluster with regard to immune infiltration. Figure 7, summarizes the total amount of cells as well as the numbers of CD45$^+$, Ly6G$^+$, and DcTRAIL-R1$^+$ cells that are found within the boundaries of the respective cluster.

Next to TMEs, we investigated the molecular heterogeneity of neutrophils. For this, the Ly6G$^+$ cells were analyzed using a UMAP and classified using Density-Based Spatial Clustering of Applications with Noise (DBSCAN) (Fig. 6c). This results in a clear separation of eight clusters. To provide linguistic distinction from clusters described for

the tissue microenvironments, results for neutrophils will be referred to as subtypes. Colored respectively and projected onto the cell map, these different subtypes can be found in distinct areas of the tumor (Fig. 6d).

Apparent from the comparison of the images depicted in Fig. 6, some subtypes show a clear spatial correlation with a specific microenvironment and histological features (Fig. 7, Supplementary Fig. 5a). Subtype 6 is almost exclusively located in the skin. Neutrophils are found in skin in comparably high number and have important and specific barrier functions[55]. Our data suggests a significantly altered lipid profile for these neutrophils. For example, in contrast to all other subtypes, subtype 6 shows a low expression of PC(O-32:0) (Supplementary Fig. 5b,c).

Neutrophils of subtypes 5 and 7 are located in specific tissue microenvironments that correlate with the adipose tissue (clusters IV and V) in the periphery of the tumor. Their lipid profiles show an increase of TG and DG lipid species (Fig. 8). Subtype 4 is predominantly found in non-adipose tissue close to the tumor (cluster VII). Similarly, subtype 2 also populates this peripheral region but is additionally found inside the tumor close to its necrotic core. Lipid profiles for both subtypes are low in DG and TG. While their overall cell density in this area is relatively high, they do not form agglomerates.

Interestingly, neutrophils in this periphery of the tumor show strong molecular similarities to their respective microenvironment (Supplementary Fig. 5c). This may be explained by an uptake of extracellular vesicles or cell debris produced by the surrounding tissue and "recycling" of the lipid content by the neutrophils. Notably, the expression of lipid species that require specific enzymatic machinery for their synthesis such as PC ether lipids (PC-O) do not show similarities to the surrounding tissue but are upregulated only in the respective neutrophil populations (Supplementary Fig. 5c). The exclusive co-localization of these ion species to Ly6G$^+$-cells also precludes the involvement of measurement artefacts, such as analyte diffusion into this convergence of lipid profiles between neutrophils and their respective TME (Fig. 5).

With the exception of subtype 4, all neutrophils found in the tumor periphery and the adipose tissue show a relatively high density of immune cells in their direct vicinity (Fig. 6a). This observation is corroborated by a neighborhood analysis that counts the average number of immune cells within a distance of 15 μm to the cell edge of each individual neutrophil (Fig. 7).

Inside the tumor, neutrophils of the subtype 1 are found mostly in the viable tumor tissue (clusters II, III, and V) (Fig. 7 and Supplementary Fig. 5a). Notably, their lipid profile shows little similarity to their respective surrounding TMEs (Supplementary Fig. 5c). This may indicate a decreased up-take and turnover of lipids or a reduced residence time within the tumor tissue. While the overall density of subtype 1 neutrophils in the tumor increases towards the adjacent adipose tissue, they are mostly isolated with relatively few other immune cells around them (Fig. 7).

In contrast, neutrophils of subtypes 3 and 8 form strong agglomerations (Fig. 6e). These subtypes are located within TME I with subtype 8 being confined to the center of the region. Their neighborhood analysis results in high numbers of overall immune cells in their direct vicinity (Fig. 7). About 90% of subtype 8 neutrophils are identified as DcTRAIL$^+$. The lipid profile of this subtype strongly differs from that of the surrounding hypoxic TME I (Supplementary Fig. 5c). It presents an increased signal for ACAR 18:0 and an increase in signal intensities for lipid metabolites, such as LPCs (Fig. 8) and DGs, while TGs are mostly absent. This may indicate an upregulation of specific lipid metabolism in this subtype.

Similar to subtype 8, neutrophils of the subtype 3 are almost exclusively found in agglomerates inside TME I but are located further away from its center. In contrast to subtype 8, however, only 17% are DcTRAIL-R1$^+$. The lipid profile of subtype 3 also contains high signal

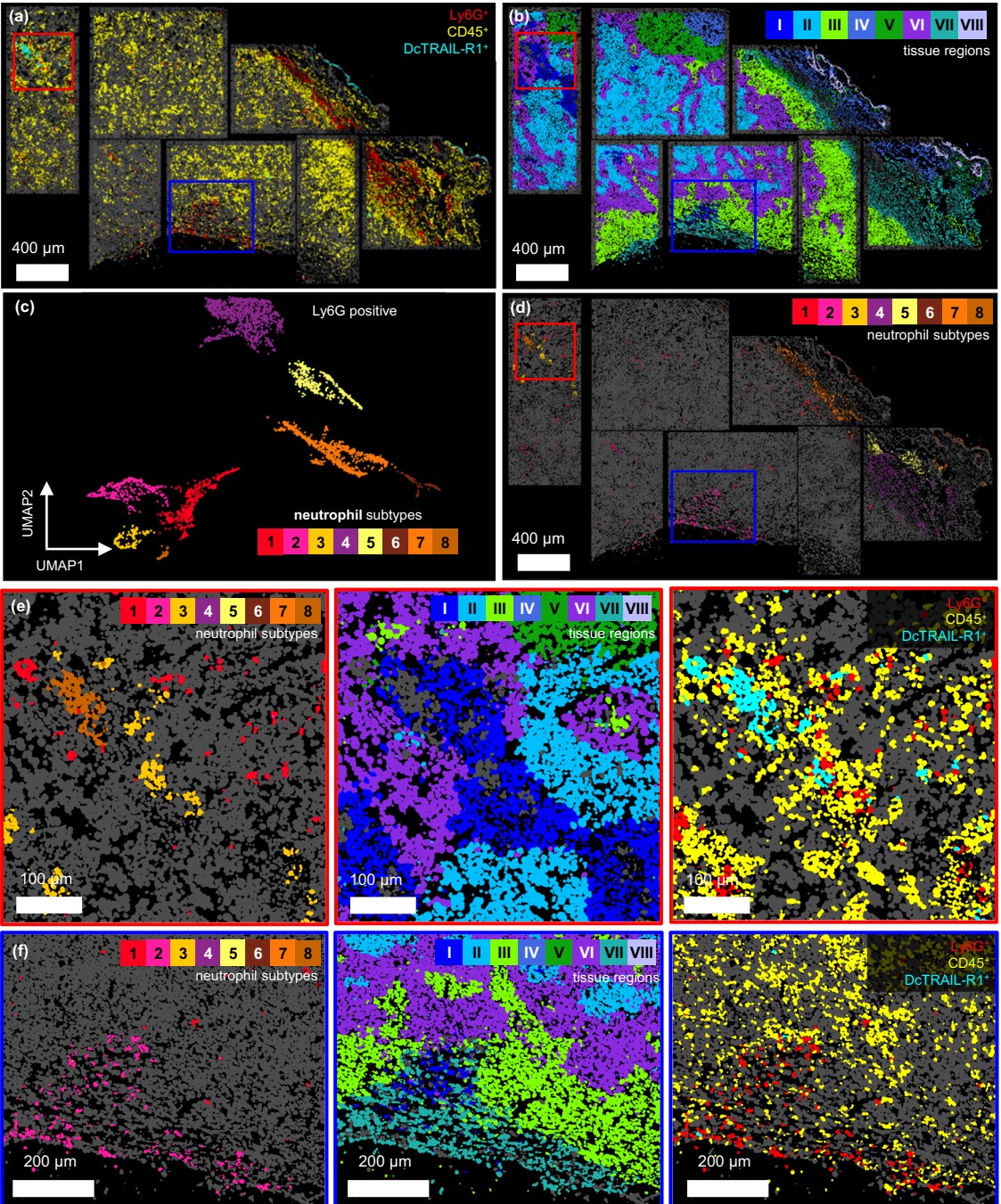

**Fig. 6 | Neutrophil lipid profile in dependence of their state and localization in the tumor tissue. a** Cell masks colored corresponding to their FM-based classification are projected onto their original position of the t-MALDI-2-MSI measurement region. Ly6G⁺ cells are depicted in red, CD45⁺ cells in yellow, and DcTRAIL-R1⁺ cells in cyan. All other cells are marked in dark gray. **b** The 59,998 Ly6G⁻ cells were separated into eight different clusters (tissue microenvironments) using k-means clustering on the standardized single-cell MS data. Cell masks for Ly6G⁻ cells are colored according to their cluster and projected onto their original position of the t-MALDI-2-MSI measurement region. All other cells are marked in dark gray. **c** 3410 Ly6G⁺ cells were identified based on FM and selected for analysis. They are plotted in a UMAP based on a list of 63 annotated ion signals. Each dot represents an individual cell, while the distance between dots visualizes the similarity of the standardized lipid profiles of the cells. Using DBSCAN, the UMAP reveals eight clearly discernable clusters (neutrophil subtypes) which are color coded. **d** Cell masks for Ly6G⁺ cells are projected onto their original position of the t-MALDI-2-MSI measurement region. All other cells are marked dark gray. The colors of cell masks correlates with the clusters found in (**c**). **e** Zoom-in of the region annotated with lymphocyte accumulation (red squares in **a**, **b**, and **d**). **f** Zoom-in of the region annotated with intratumoral hemorrhage (blue squares in **a**, **b**, and **d**).

intensities for ACAR 18:0 but reduced levels of LPC and DG. Again, in contrast to subtype 8, the lipid profile of subtype 3 neutrophils shows great similarities to its surrounding TME I. This increased molecular crosstalk may indicate a longer residence time of neutrophils within this hypoxic TME as opposed to the viable TMEs II, III, and VI.

## Discussion

In its own right, MSI analysis can provide specific molecular information of single cells, thereby delivering an important puzzle piece towards the understanding of individual cellular behavior. For a more holistic approach, however, the resulting lipidomic or metabolomic

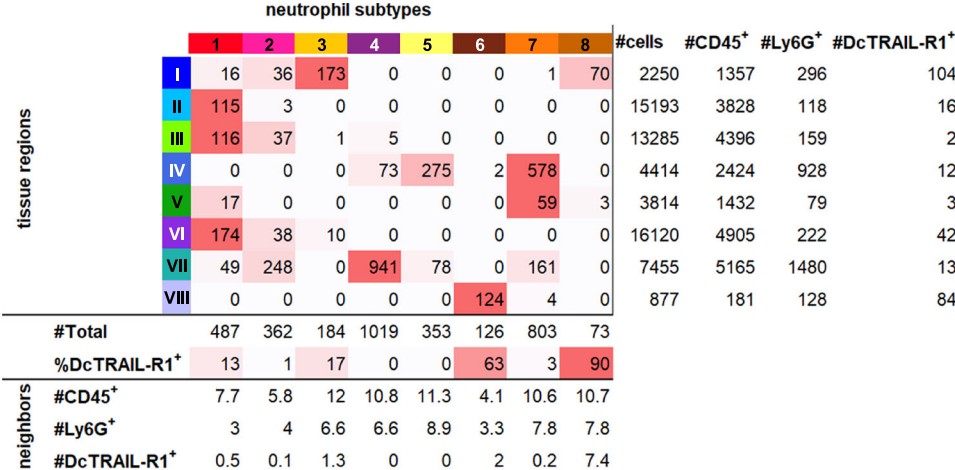

| | | neutrophil subtypes | | | | | | | | #cells | #CD45+ | #Ly6G+ | #DcTRAIL-R1+ |
|---|---|---|---|---|---|---|---|---|---|---|---|---|---|
| | | 1 | 2 | 3 | 4 | 5 | 6 | 7 | 8 | | | | |
| tissue regions | I | 16 | 36 | 173 | 0 | 0 | 0 | 1 | 70 | 2250 | 1357 | 296 | 104 |
| | II | 115 | 3 | 0 | 0 | 0 | 0 | 0 | 0 | 15193 | 3828 | 118 | 16 |
| | III | 116 | 37 | 1 | 5 | 0 | 0 | 0 | 0 | 13285 | 4396 | 159 | 2 |
| | IV | 0 | 0 | 0 | 73 | 275 | 2 | 578 | 0 | 4414 | 2424 | 928 | 12 |
| | V | 17 | 0 | 0 | 0 | 0 | 0 | 59 | 3 | 3814 | 1432 | 79 | 3 |
| | VI | 174 | 38 | 10 | 0 | 0 | 0 | 0 | 0 | 16120 | 4905 | 222 | 42 |
| | VII | 49 | 248 | 0 | 941 | 78 | 0 | 161 | 0 | 7455 | 5165 | 1480 | 13 |
| | VIII | 0 | 0 | 0 | 0 | 0 | 124 | 4 | 0 | 877 | 181 | 128 | 84 |
| | #Total | 487 | 362 | 184 | 1019 | 353 | 126 | 803 | 73 | | | | |
| | %DcTRAIL-R1+ | 13 | 1 | 17 | 0 | 0 | 63 | 3 | 90 | | | | |
| neighbors | #CD45+ | 7.7 | 5.8 | 12 | 10.8 | 11.3 | 4.1 | 10.6 | 10.7 | | | | |
| | #Ly6G+ | 3 | 4 | 6.6 | 6.6 | 8.9 | 3.3 | 7.8 | 7.8 | | | | |
| | #DcTRAIL-R1+ | 0.5 | 0.1 | 1.3 | 0 | 0 | 2 | 0.2 | 7.4 | | | | |

**Fig. 7 | Data matrix of neutrophil subtypes across tumor microenvironments.** Data matrix tabulating the number of neutrophil subtypes found in each tissue region. For each subtype, the percentage of DcTRAIL-R1+ cells is calculated. In a neighborhood analysis for all neutrophils, the mean number of cells categorized as CD45+ cells, Ly6G+ cells and DcTRAIL-R1+ cells found in a 15 μm radius are presented. Additionally, for each microenvironment, the total number of contained cells, CD45+, Ly6G+ and DcTRAIL-R1+ is noted.

information has to be put into the context of the cell type, differentiation status, and state, but also has to regard its current tissue macro- and microenvironment[1]. Here, a tight connection to optical microscopy analysis has long since been identified as a key feature in the integration of MALDI-MSI into the "biomedical toolbox"[56,57]. In this context, high-quality microscopy analysis, including IF, is typically carried out on adjacent tissue sections, introducing well-known challenges in the accuracy of co-registration. Because of their small size in the range of section thickness, this particularly limits the analysis of specifically targeted single cells[33,56,58,59]. For the microscopic analysis of the same section, staining-based microscopy is conventionally carried out after MALDI-MSI. While impressive progress has been reported very recently in this field, damage or distortion of the tissue introduced during the measurement and the removal of the matrix can impede accurate co-registration on the cellular level, especially for large areas of interest. In addition, cross-reactions with the matrix and laser irradiation that precedes microscopy, can impede binding properties for antibody based analysis[12,60–63]. Successful microscopy of the same sample prior to MALDI-MSI has largely been restricted to the limited analytical capabilities offered by unstained tissue, e.g. by autofluorescence or phase-contrast microscopy[27,33,64,65]. Alternatively, other non-destructive optical methods, such as IR-imaging and micro-Raman imaging, have been employed prior to MSI analysis but have not been utilized for automated cell segmentation[66,67].

Exceeding these current limitations, the technique presented here includes slide scanning microscopy outside and inside the ion source of the same sample recorded prior to ultra-high resolving power t-MALDI-2-MSI without compromising performance in either modality. Of note this includes the successful implementation of pre-MALDI IF analysis.

Dedicated staining techniques allow for a robust and automated cell segmentation and retain the full spectral information in MSI. Uniquely, the high fidelity co-registration of both modalities for a large field of view facilitated by in-source FM that omits the need of "cross-modality" fiducial markers, allows for the automated generation of single-cell mass spectra from tissue in large numbers. While previous efforts that relied on manual steps in co-registration and definition of cellular ROIs[21,68], this automation lays the groundwork for the reliable and large-scale, generation of information-rich highly resolved data for both modalities on the same tissue sample.

Two exploratory studies highlight the advantages of the described method for the molecular analysis with cellular and even sub-

cellular resolution. Using the phagocytosis of *E. coli* bioparticles by macrophages as an example, we demonstrate how the technique can be used to investigate metabolic processes inside sub-cellular compartments like the phagolysosomes. Assignment of bacterial PE 33:1 and PE 34:2 confirmed the uptake of *E.coli* bioparticles. The additional identification of increased amounts of the respective DG species suggests their partial degradation. In addition, the tentative identification of several LPG species provides further useful leads to the metabolic breakdown of *E. coli* derived CL and PG by enzymatic degradation and/or oxidation of lipids inside the phagolysosome[69].

Overall, this study illustrates a dedicated approach to analyzing molecular profiles of subcellular compartments. In future applications, this largely untargeted technique may be used to investigate and monitor metabolic turnover on the organelle level of specific single cells.

In our study of the 4T1 tumor model, we demonstrate spatial biology analysis on the lipid level for highly heterogeneous tissue. While previous work on single-cell MSI from tissue sections has been limited by comparatively large pixel size and by the manual annotation of small numbers of cells[17–19,70], automated segmentation based on slide scanning (immuno-)FM resulted in the identification of more than 63 K cells and allowed for the identification and classification of common immune cells (CD45+, ~ 23.7 K cells; 38% of all cells), relatively rare neutrophils (Ly6G+, ~ 3.4 K cells; 5.4%), and very rare tumor associated T3-neutrophils (DcTRAIL-R1+, 276 cells; 0.4%) within the tissue. Importantly, the segmentation, in combination with highly accurate co-registration of the microscopy with t-MALDI-2-MSI results, allowed for the generation of single-cell mass spectra for each cell.

Using statistical analysis, cells were clustered based on their lipid profile, revealing eight distinct tissue microenvironments inside the tumor and around its periphery. Likewise, an independent clustering of just the Ly6G+ cells revealed eight subtypes of neutrophils inside the investigated tissue region. Comparison of lipid profiles between neutrophils and their direct tissue environment revealed striking similarities in some cases. These results allow for a new perspective on the uptake and recycling of lipids by specific neutrophil subtypes. They also reveal distinctive differences between neutrophils and their direct environment, such as the increased levels of ether lipid species, also described in the literature[49]. In addition to a general upregulation of these lipid species, we are able to identify subtype and TME-specific differences in their expression. Neighborhood analysis of all investigated immune cells in the direct vicinity of the neutrophils provides a

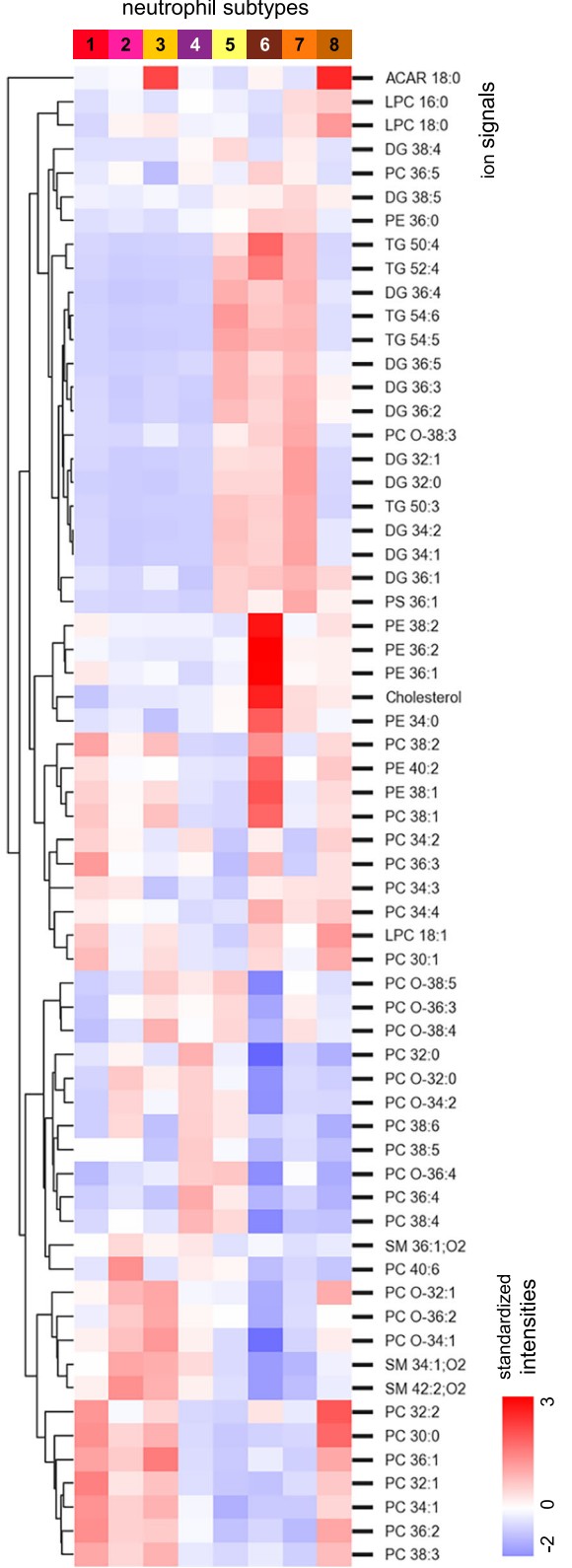

**Fig. 8 | Heatmap of lipid profiles across neutrophil subtypes.** A heat map of standardized intensities of 63 ion species for the eight neutrophil subtypes visualizes differences in lipid profiles.

first hint at their swarming behavior. While some of the subtypes represent relatively isolated cells, others are found in agglomeration with neutrophils and other immune cells.

The strength of the molecular investigation at the cellular level becomes especially apparent in the examination of the hypoxic regions of the tumor. This TME shows a signal intensity increase for ACAR 18:0, as well as a high density of immune cells, both described as markers for hypoxia[51]. The inner core of this TME is populated by densely packed T3-neutrophils (most of them of subtype 8), concurring with the literature[46,47]. A second group of agglomerated mostly DcTRAIL-R1⁻ neutrophils (subtype 3) is also located inside the same hypoxic area but further away from its center. The subtyping now enables an in-depth analysis of underlying lipid profiles and integration with the state of research. While both subtypes, for example, express increased levels of ACAR 18:0, only subtype 8 (mainly DcTRAIL-R1⁺ cells) shows increased levels of fully saturated LPCs. These mediator are known to be important in metastasis formation and in the recruitment of phagocytes during apoptosis[71,72] and our data points to a previously unknown tentative connection with T3-neutrophils.

While a full investigation of the underlying biological mechanisms is not within the focus of this publication, the presented proof-of-concept studies showcase the great potential of the technique. The seamless integration of morphometric data with (sub-)cellular lipid and possibly metabolite analysis significantly expands the informational space available for each individual cell inside the tissue. Exceeding the current state-of-the art that is largely limited to the analysis of cellular neighborhoods, our approach enables a direct in situ contextualization of single individual cells with their direct tissue microenvironment on the combined molecular and morphometric level.

The presented techniques introduces a versatile set of tools for an improved integration of single-cell MSI into existing workflows. Embedded within a full suite of spatial biology methods, in future applications, it will allow for the combination of IF-based information that may describe the regulation of specific genes and the expression of individual proteins with the expression of lipids and possibly also metabolites in the same cell and its direct surrounding[73]. This addition of "spatial biology of lipids and metabolites" on the single-cell level provides an important additional layer of information to complete pathway analysis from the transcript and proteome level down to the regulation of the molecular products. It can be a valuable asset to review and validate downstream effects of genetic or transcriptomic variation, as well as protein inhibition, but also help to investigate regulatory functions of small molecules on gene and protein expression. Future developments in t-MALDI-MSI will need to focus on further increasing the sensitivity and analytical depth of the method. E.g., innovative approaches to sample preparation combined with plasma-based ionization approaches, as described by Young et al. in the same issue of this journal, can help to increase sensitivity and assist in the analysis of additional classes of metabolites[74].

Overall, the methods described here have great potential to help untangle the influence of endogenous and exogenous factors such as immune cell infiltration, infection, or drug delivery on cellular behavior in a specially resolved manner. They are, however, by no means limited to these applications but can be applied to decipher open biological questions in a wide range of tissues and cell types.

## Methods
### Reagents, resources, and software
All reagents, resources, and software used are listed in the table of resources (Supplementary Table 4)

### Cell culture preparation
THP-1 ACC-16 cells (DSMZ) were cultured in T25 cell culture flasks at 37 °C and 5% $CO_2$ in RPMI 1640 Medium (Lonza), which was supplemented with 2 mM L-glutamine, 10% fetal bovine serum and 1 mM sodium pyruvate. About 10,000 cells per chamber were sown in

Millicell EZ Slide 8-well chamber slides (Sigma-Aldrich). Differentiation to M0 macrophages was induced by exposure to 20 ng/ml phorbol 12-myristate 13-acetate (PMA, Sigma-Aldrich) for 24 h followed by a 3-day rest phase in PMA-free medium. Two hours before harvest, pHrodo™ Red *E. coli* BioParticles (Thermo Fisher) were added to the culture medium resulting in a concentration of 95 µg/mL. Cells were washed with PBS and fixed for 5 min using 4% formaldehyde (low-methanol, Roth). F-Actin and nuclei were stained according to a protocol described by Bien et al. using 1X CellMask Green Actin Tracking Stain (Thermo Fisher) diluted in PBS for 30 min and 1 µg/mL Hoechst 33342 (Sigma-Aldrich) dissolved in PBS for 5 min[17]. Cells were washed with PBS after every staining step. After staining, cells were washed using 150 mM ammonium acetate (Sigma-Aldrich) and dried at room temperature for 2 h.

## Mouse model

All animal experiments were carried out in accordance with local animal welfare guidelines approved by the responsible authorities (North Rhine-Westphalia State Agency for Nature, Environment and Consumer Protection, LANUV, Protocol No. 81-02.04.2018.A010). Female BALB/c mice (Charles River Laboratories) were used at the age of 8–12 weeks. Mice were housed under a 12 h light/dark cycle with ad libitum access to food and water. For tumor implantation, $10^6$ 4T1 cells were resuspended in 25 µl of DMEM cell culture medium (Thermo Fisher) and then injected orthotopically into the lower left mammary fat pad of the mice. Tumors were allowed to grow for 9 days and tumor size was monitored daily with a digital caliper. The local ethics committee approved a maximum tumor diameter of 8 mm. This limit was strictly adhered to and not exceeded in all experiments conducted within this study. On day 9 after tumor implantation, the animals were sacrificed. Brain tissue was harvested from untreated mice used as the control group for the mentioned study.

## Tissue sectioning

Mouse brain and tumor tissue were embedded in Epredia M-1 Embedding Matrix (Thermo Fisher) and snap-frozen in liquid nitrogen. 10 µm (cerebellum) or 8 µm (tumor) thick tissue sections were produced with a cryostat (CM 3050 S, Leica Biosystems) at −20 °C. Sections were thaw-mounted onto SuperFrost glass slides (Thermo Fisher) for H&E-staining or IntelliSlides (Bruker) for t-MALDI-2 MSI measurements. Samples for t-MALDI-2 MSI were vacuum-sealed and stored at −80 °C before further use.

## Tissue fixation and fluorescence staining

Frozen tissue sections were slowly brought to room temperature in the vacuum-sealed package. After opening the seal, sections were further dried under a gentle nitrogen stream for 10 min. For unstained control tissue the section was directly transferred to the sublimation chamber. For staining, a contour was drawn around each section using a Super PAP Pen (Science Services), followed by another 10 min drying under a gentle nitrogen stream, to create a hydrophobic barrier and thus enabling pipetting aqueous solutions on and off the sample. The following washing, fixation, and staining steps were all performed by pipetting ~100 µl of the accordant solution onto the sample and removing it after the specified time. Slides with tissue sections were placed on a metal plate standing in ice water. Samples were quickly fixed in 4 % formaldehyde (low-methanol, Sigma-Aldrich) in PBS for 5 min and subsequently washed with PBS for 30 s. After tissue fixation, the primary antibody, Ly6G (ab307167, Abcam), DcTRAIL-R1 (130-110-873, Miltenyi Biotec), CD45 (130-110-798, Miltenyi Biotec) or Anti-Calbindin (ab229915, Abcam), diluted 1:500 (v/v) in 1% BSA (Serva) in PBS was applied to the sample and incubated for 60 min. The sample was then washed three times for 30 s in PBS. In case of a dual antibody staining, the secondary antibody, Alexa Fluor 594 (ab307167, Abcam), diluted 1:1000 (v/v) in 1% BSA in PBS was applied and incubated for 60 min, followed by three times 30 s washing steps in PBS. 5 µg/ml Hoechst 33342 (Sigma-Aldrich) and 1X CellMask Green Actin Tracking Stain (Thermo Fisher) diluted in PBS were applied and incubated for 5 min. After staining, the slide was lifted from the metal plate and parts of the hydrophobic barrier were removed at the bottom of the slide with a pipette tip. Samples were then washed with 2 ml of ammonium acetate (Sigma-Aldrich) each, by letting it flow carefully over the section. For final drying, samples were placed under a gentle $N_2$ stream for 20 min.

## Tissue H&E staining

Hematoxylin and eosin (H&E) stains of tissue sections were performed on untreated consecutive sections directly after sectioning. Sections were stained for 30 s with hematoxylin (Sigma-Aldrich) with subsequent bluing in tap water for 2 min. Afterwards, the tissue was stained for 30 s with eosin (Sigma-Aldrich) and then rinsed with water.

## External microscopy

Fluorescence images of cell cultures and tissue sections were recorded at 50X using the DAPI, FITC, TxRed, Cy3, and Cy5 channels of a slide scanning microscope (VS200, Evident) equipped with a SpectraSplit 7 filterset (Kromnigon). For each channel individually, the exposure time was set by performing the auto exposure procedure of the microscope on a region with high signal intensity for the respective fluorophore. This procedure was conducted prior to the imaging run, and the entire slide was subsequently imaged with a constant exposure time, with a maximum exposure of 500 ms. H&E-stained tissues were recorded at 20X in BF mode. Image processing was performed with OlyVIA (Evident). Channel histograms were linearly adjusted for display. Channels are shown with linear LUTs (red, green, blue, cyan, magenta, yellow). For further information please refer to the light microscopy reporting table (Supplementary Table 5).

## Transmission MALDI optics

A schematic of the optical set-up used for material ejection for the t-MALDI-2-MSI analysis and in-source brightfield and FM image generation is depicted in Fig. 1a-c of the main article. The sample is mounted on a three-axis piezo stage (SmarAct) and is illuminated homogeneously by 15 green LEDs (LED528EHP, Thorlabs) mounted on top of a 50X objective (M Plan Apo NUV HR 50X, Mitutoyo) with a custom-designed 3D printed LED ring. Light from the sample is collected by the objective and focused by a tube lens (TTL100-A, Thorlabs) onto the sensor of a digital camera (XCG-CG160, Sony). A simple top-view observation of the sample is in addition possible by the pre-installed camera of the prototype timsTOF fleX MALDI-2 mass spectrometer.

The beam path of the MALDI laser (smartbeam, wavelength 355 nm, pulse duration 7 ns) of the timsTOF fleX MALDI-2 mass spectrometer is re-routed and condensed through a 6:1 telescope optic and co-aligned to the microscopy path with a dichroic mirror (L-07246, Laseroptik Garbsen). For recording FM images, a filter cube (DFM2/M, Thorlabs) with a filter set (DFM2T1, Thorlabs) for the suitable fluorescence channel (MDF-BFP, DAPI; MDF-FITC, FITC; MDF-MCHC, mCherry; all Thorlabs) consisting of excitation and emission filters and a dichroic mirror is mounted in front of the tube lens. In fluorescence mode, the LED ring is turned off and fluorescence is excited by the respective fiber-coupled LED (M385FP1, DAPI; M455F3, FITC; MINTF4, mCherry; all Thorlabs) mounted with a collimating optic directly to the filter cube. The camera is integrated into the software by using the XC-SDK 2018 (Sony). The trigger signal for image acquisition of the camera and fluorescence excitation is provided as a "position reached" event by the controller of the piezo stage and is handled by a digital delay pulse generator (9200 Sapphire Series, Quantum Composers). With this pulse generator, the fluorescence excitation time was set from

10 ms to 2 s per image depending on the sample and fluorescence channel.

## MALDI-matrix coating by resublimation

To achieve a reproducible and homogeneous coating of the sample slides with MALDI-matrix, a custom-designed resublimation chamber was developed. A schematic of this setup is shown in Supplementary Fig. 6a, b. Under atmospheric pressure conditions, the hot-side is preheated to 80 °C and 0.5 ml of 10 mg/ml CHCA MALDI-matrix (Sigma-Aldrich) in acetone solution is pipetted into the matrix-reservoir. With the acetone evaporating rapidly, the MALDI-matrix is left forming a uniform layer on the bottom of the matrix reservoir. The sample is then clamped onto the cold-side at room temperature and the whole device is sealed and evacuated. After reaching a final pressure of $10^{-2}$ mbar, the cold-side is first cooled to -18 °C, then the hot side is heated to 150 °C to start the sublimation and resublimation. After 60 min, the heating of the hot-side is turned off and the cold-side is heated to room temperature before venting the system to prevent condensation of ambient humidity on the sample. The sample is then put onto a room temperature metal block in a saturated atmosphere of 0.5% ethanol in water at 70 °C for 1.5 min for recrystallization. To demonstrate differences in contrast for in-source BF microscopy induced by the choice of matrix, cerebellum samples were coated with DHB and HABA MALDI-matrix (both Sigma-Aldrich) using the same settings as described above for CHCA (Supplementary Fig. 6c).

## Autofocus of optics

The MALDI laser and microscope camera utilize the same objective for focusing and observation in the transmission-MALDI setup employed in this work. A schematic of the setup is depicted in Fig. 1a in the main article. The focal plane of the MALDI-laser is adjusted to match that of the camera by fine adjusting the second telescope lens of the laser beam path. After aligning the focal planes, all optics are kept at a constant position. The objective (M Plan Apo NUV HR 50X, Mitutoyo) has a small focal depth of 1.4 μm and samples have to be in focus during optical microscopy and MALDI-MSI. This can be achieved by methodically adjusting the z-position of the sample with regard to the objective while recording sequential optical microscope images at z-increments of 1 μm. For each of these images a Laplacian filter is applied and the variance of the result is calculated, which serves as a measure of image sharpness. The stage is then moved to the z-position corresponding the maximum sharpness image and a fine-scan with 0.1 μm z-distance is performed in the range of +/- 1 μm in the same way as before to find the z-position of highest sharpness at a submicron precision. Since the focal planes of the optical microscope and the MALDI laser are aligned, this optical focusing ensures an optimal focus of the MALDI-laser on the sample. This autofocus routine can be executed both BF and FM mode. The Python code of this program Auto-FocusClass.py is provided in Data S1 in the supplementary information.

## In-source microscopy workflow

The scale factor to convert pixel on the camera sensor to position of the stage was calculated by using a 1951 USAF Resolution Test Target (Thorlabs). For selection of a region of interest (ROI) for an optical microscope scan in the ion source, autofocusing points are placed manually on the sample. The ROI is defined as the smallest enveloping rectangle that contains all points. The stage is then moved to each of the autofocusing points and an autofocus routine as described in "Autofocus of optics" is performed, which yields the focal z-offset of that xy-coordinate. Using a radial basis function interpolation, the z-offset is calculated for every coordinate in a 10 x 10 μm mesh inside the ROI and saved to a z-interpolation file. After creating the z-interpolation, the optical microscope image of the ROI is taken. For this an up to 100 μm x 100 μm wide x/y-position mesh is created in the ROI. An optical microscope image is then taken at each mesh point at the z-offset of the closest point to the xy-position from the z-interpolation file. To create a complete image of the ROI, an empty canvas of the correspondent size is created. Using the scale factor of the setup, images are cropped from the edges to a size of 100 x 100 μm and are placed onto the empty canvas using the center coordinates from the xy-position mesh. After creating a z-interpolation file for a ROI once, multiple modalities, such as brightfield or different fluorescence channels can be recorded without creating a new interpolation. ImageJ was used to create overlay images of different modalities. The Python code of this workflow can be found in the files RecordPositions.py, ZforRecordPositions.py, Scanning.py and stitching.py in Data S1 in the supplementary information.

## Transmission-MALDI-2 MSI

All experiments were carried out in positive ion mode with the exception of one measurement from a brain tissue section. The MALDI-MSI run is set up partly using the proprietary flexImaging software (Bruker). Instead of using an external scan of the sample and teaching that to the stage coordinate system, the (multimodal) image of the ROI (see "In-source microscopy workflow") is used. In the classic flexImaging workflow, three teaching points would be manually selected on the optical image. The corresponding stage coordinates would then be recorded by driving the sample-stage manually to the respective positions. Since center coordinates and pixel positions on the ROI image are already established from recording the ROI image, three x/y-positions from the 100 x 100 μm mesh (top left, bottom left, bottom right) and their corresponding stage coordinates are automatically deployed as teaching points. With these points, the setup file for the t-MALDI-2 MSI run can be automatically created, defeating any positioning errors introduced by manually selecting teaching points in the classic workflow. When the MALDI-laser is not directed at the exact middle pixel of the in-source optical microscope, fine adjustments of a few pixels offset can be applied at this stage. The automatically created setup file for the t-MALDI-2 MSI run can be opened directly in flex-Imaging to create measurement regions, select settings like pixel size, and laser shots per pixel. The previously acquired z-interpolation is recognized by timsControl software during t-MALDI-2 MSI measurements by providing the interpolation mesh as a geometry file. The Python code for creating the flexImaging setup files can be found in Createmis.py Data S1 in the supplementary information. All t-MALDI-2 MSI experiments were performed with 25 laser shots per pixel and a pixel size of $1 \times 1\ \mu m^2$ for the cerebellum and tumor sample and $1.5 \times 1.5\ \mu m^2$ for cell culture samples. Signal intensity distributions are visualized using SCiLS Lab software (Bruker) using weak de-noising.

## Ion signal annotation

For the THP-1 cells lipid signals were identified as described by Schwenzfeier et al. using shotgun lipidomics by electrospray ionization (ESI) mass spectrometry on a Q-Exactive Plus orbitrap mass spectrometer (Thermo Fisher) equipped with a dual ion funnel[20,75]. Lipid extracts of 72 h PMA stimulated THP-1 macrophages were produced by an adapted protocol of Folch et al. [76]. Approximately $10^6$ cells were centrifuged at 1000 g to separate them from the medium. Cells were resuspended in 1 ml $CHCl_3$/MeOH (2:1, v/v) and lyzed for 10 min in an ultrasonic bath at room temperature followed by an incubation of 50 min at 4 °C in a cryoshaker at 500 RPM. Phase separation was induced by addition of 240 μl $H_2O$ and centrifugation at 9880 g for 10 min. The organic phase was transferred to a glass vial and the solvent was removed using a nitrogen evaporator. Lipid extracts were vacuum sealed and stored at -80 °C. For ESI-MS analysis, extracts were re-dissolved in 500 μl MeOH and measured both in positive and negative ion modes using a flow rate of 5 μl/min. A data dependent analysis workflow (DDA) was applied using tandem MS based on collision-induced dissociation (CID) with stepped normalization collision energies (NCE) of 20, 25, 30 and an exclusion time of 300 s. The

mass resolving power was set to 280,000 (at *m/z* 200) with an AGC target of 1E6 and a maximum injection time of 250 ms. The resulting data was merged with a t-MALDI-2-MSI dataset and imported into LipostarMSI (Molecular Horizons) for annotation. The resulting feature list was hand-curated, which involved removing of isotope peaks and addition of known species from *E. coli*.

For mouse cerebellum and the 4T1 murine tumor section, annotations were based on accurate mass (< 3 ppm). For this, signals with S/N > 3 were extracted from the mean spectrum and annotated using the LipidMaps "Bulk Structure Search"[77]. In positive ion-mode sphingolipids, glycerolipids, sterol lipids and glycerophospholipids were selected as possible lipid classes either as $[M + H]^{+}$- or $[M-H_2O + H]^{+}$-ions. The comparative analysis of untreated tissue also included $[M +Na]^{+}$ and $[M + K]^{+}$ ion species. In negative ion-mode only $[M-H]^{-}$-ions were searched for. The resulting annotation lists were manually curated to remove biologically unlikely annotation (e.g., lipids with odd side chains) or lipid classes that are usually not detected in the respective ion-mode (e.g., glycerophosphoinositols in positive ion-mode). Using this method, a total of 66 ion signals for the mouse cerebellum and 63 ion signals for the 4T1 tumor sample were annotated, respectively.

The lipid annotation of mouse cerebellum was subsequently matched with data from a full lipidomics analysis reported in the literature[32]. Because t-MALDI-2-MSI without the use of additional techniques like OzID or Paternò-Büchi-reactions[78,79] is not able to differentiate isomers within a lipid class (e.g., PC(16:0/18:1) and PC(16:1/18:0)) or between lipid classes (e.g., PE(16:0/18:0) and PC(15:0/16:0)) the literature data was reduced to 255 unique lipid species according to the here employed level of specificity. Of the 66 annotated *m/z*-values, 63 were also detected by Fitzner et al. [32]. The remaining unmatched values primarily corresponded to sulfoglycosphingolipids and fatty acids, lipid classes that were not analyzed in the original publication. Overall, the annotation accounted for approximately 82% of the total lipid mass identified by Fitzner et al., calculated by summing all lipids found in both datasets and using the quantitative amounts reported.

Lipid nomenclature and shorthand notation is reported according to Liebisch et al. [80].

### Generation of single-cell mass spectra and spectra of subcellular features

The external and in-source fluorescence images were co-registered using SimpleITK 2.3.1 (NumFOCUS)[81,82]. For initialization a landmark registration was used. Based on the resulting translation matrix an affine image registration was performed using Mattes mutual information as metric and the gradient descent algorithm as optimizer[83]. The number of bins for the mutual information was set to 50, the parameters for the optimizer were set as follows: learning rate (1), number of iterations (50), convergence minimum value (1E-6) and convergence window (20). The sampling strategy was set to random and 1% of the pixels were sampled. The resulting transformation matrix was combined with the initial translation matrix to result in a global transformation matrix. Due to the inherent link between the t-MALDI-2-MSI measurement and the in-source fluorescence images, the global transformation matrix could be used to resample the external fluorescence images to fit the exact region of the t-MALDI-2-MSI measurement. The precision of the registration was determined by placing pairs of landmarks at distinct features, which are visible in the external fluorescence image as well as the MALDI image of the cerebellum dataset (Supplementary Fig. 1b). While the precision of the placement of each individual landmark is limited by the manual selection process and differences between modalities, overall accuracy is ensured by determining the average over a total of eleven pairs of landmarks. The target registration error (TRE) was then determined by projecting the landmarks of the MALDI image into the coordinate system of the

external fluorescence image using the global transformation matrix and calculating the distance to the corresponding landmarks of the external fluorescence image (Supplementary Fig. 1c and Supplementary Note 1). Cell masks for tumor tissue were created by using Deep-Cell Mesmer 0.12.9 (Van Valen Lab)[53] based on the DAPI channel (Hoechst 33342, nuclei) and the FITC channel (CellMask Green Actin Tracking Stain, actin). Outlines for subcellular features were determined by thresholding fluorescence signal intensity. Using FISCAS[20], mass spectra were calculated for each cell or subcellular compartment by compiling MSI pixels based on their co-localized masks. MSI pixels assigned to multiple cell masks were scaled according to the hit coverage of the segmentation masks. Morphometric parameters were acquired using CellProfilers 4.2.1 (Cimini Lab)[54] measurement modules.

### Cell classification

For cell classification, immune cells were identified and divided into three groups (CD45$^+$, LyG6$^+$, DcTRAIL-R1$^+$) based on thresholding the mean intensities of the corresponding antibody stains in their respective fluorescence channels (Cy5, TxRed, Cy7). Ly6G$^+$ cells showed high intensities on the TxRed and Cy5 channel, while DcTRAIL-R1$^+$ cells showed high intensities on all three channels. The clustering of the tumor microenvironment (TME) was done using k-means clustering on the standardized MS data of all Ly6G$^-$ cells. The number of clusters was determined by visual inspection of the clustering results for values of k between 6 and 10. The Ly6G$^+$ cells were also clustered using Density-Based Spatial Clustering of Applications with Noise (DBSCAN) with an epsilon of 0.7 and a minimum number of seven neighbors, based on a uniform manifold approximation and projection (UMAP) of their standardized mass spectra. The clusters are visualized both directly on the cell masks as well as in a UMAP of the standardized mass spectra using default parameters. Ly6G$^+$ cells were assigned to a specific tumor region, by identifying the most frequent TME label of all neighbors within 40 μm.

### Statistics & reproducibility

The presented work describes method development and can be considered proof-of-concept. No statistical method was used to predetermine sample size. For cerebellum and tumor tissues, multiple technical replicates with varying staining and sample preparation conditions were performed, and representative datasets were selected for display. For both tissue types, two additional MSI experiments with optimized sample preparation but varying fluorescence stains yielded comparable results. For the tumor sample, the full staining panel was applied once. For cell experiments, two independent replicates with consistent outcomes were performed, and representative data are shown. No data were excluded from the analyses. The experiments were not randomized, and the investigators were not blinded to allocation during experiments and outcome assessment.

### Ethics statement

The research presented here complies with all ethical regulations and was approved by the local ethics committee of the University of Münster and the North Rhine-Westphalia State Agency for Nature, Environment and Consumer Protection, LANUV, where applicable.

### Reporting summary

Further information on research design is available in the Nature Portfolio Reporting Summary linked to this article.

## Data availability

MALDI MSI data, fluorescence and brightfield microscopy images as well as processed data have been deposited at OMERO and are publicly available including an interactive data viewer at https://doi.org/10.57860/min_prj_000012[84]. Any additional information required to open

or reanalyze the data reported in this paper is available from the lead contact upon request. Source data are provided with this paper.

## Code availability

All original code has been deposited at OMERO and is publicly available at https://doi.org/10.57860/min_prj_000012. The code is included in the main folder (Data S1.zip) and within the tumor project (Data_tumor.zip) and additional information is available from the respective readme files. Any additional information required to use code reported in this paper is available from the lead contact upon request.

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

## Acknowledgements

We thank Dagmar Mense for help with cell culture, Mathis Richter for helpful discussions on immunohistochemistry, Alexander Fengler and Vincent Suerdieck (all University of Münster) for help with mouse tissue sectioning, Benedikt Geier (Stanford), Robert Ernst and Hyojik Yang (both University of Maryland) for helpful discussions, Henning Peise, Arne Fütterer, Andreas Wöste, Raik Milautzki (all Bruker) for hard- and software support, and Sarah Weischer, Jens Wendt, and Thomas Zobel (Münster Imaging Network Cells in Motion Interfaculty Center) for support with image analysis. Figure 1 was created in part using biorender (agreement number ZR28JUV8U0). Financial support from the German Research Foundation to KD and OS (Z1 project of the TRR332; project no. 449437943) and to JS (project no. 544444139) is gratefully acknowledged.

## Author contributions

Conceptualization: K.D., J.H., M.N., A.P., and J.S.; Data curation: S.B., A.P., J.Sch., and J.S.; Formal analysis: S.B., A.P., and J.Sch.; Funding acquisition: K.D., J.H., O.S., and J.S.; Investigation: S.B., M.N., A.P., J.Sch., and J.S.; Methodology: S.B., M.N., A.P., J.Sch., and J.S.; Project administration: K.D. and J.S.; Resources: K.D., J.H., E.H., O.S., and J.S.; Software: A.P. and J.Sch.; Supervision: K.D., O.S., and J.S.; Validation: K.D., J.H., and O.S.; Visualization: S.B., A.P., J.Sch., and J.S.; Writing – original draft: A.P. and J.S.; Writing – review & editing: S.B., K.D., J.H., E.H., M.N., A.P., J.Sch., O.S., and J.S.

## Funding

## Competing interests

The authors declare the following competing interest(s): MN and JH are employees of Bruker Daltonik GmbH & Co KG (Bremen). KD, MN, AP, and JS have patents pending regarding parts of the hardware development (WO/2024/041681, WO/2025/067576). O.S. receives funds from Novo Nordisk and consults Novo Nordisk and Roche. Parts of this research have been funded by Bruker Daltonik GmbH & Co KG by supplying hardware, software, and salaries for AP and JSch. The remaining authors declare no competing interests.
