## [Transparent Peer Review file · Nature Communications]

Spatial biology using single-cell mass spectrometry imaging and integrated microscopy

Corresponding Author: Dr Jens Soltwisch

Version 0:

Reviewer comments:

Reviewer #1

(Remarks to the Author)

The authors Potthoff et al. present a manuscript entitled “Spatial biology using single-cell mass spectrometry imaging and integrated microscopy” in which they present a prototype transmission mode MALDI-2 ion source configuration that also allows for in-source bright field (BF) and fluorescence microscopy (FM) to be performed on a single sample, enabling coupled (sub-)cellular analysis using both the MSI and microscopic methods. This configuration represents a further step forward in terms of the instrumental developments presented by the research team in question and, in particular, responds to a rather evident need in the field related to challenges faced when performing multi-modal imaging and the precise co-registration needed in order to arrive at the single and, in particular, sub-cellular levels.

On the whole, the manuscript is written in a thorough manner and appears to be technically sound. Moreover, it seeks to address a key issue that hinders the further progress of this technology within the context of spatial biology, especially at the single-cell level, and therefore represents noteworthy research. Notwithstanding, there are some minor concerns that should be addressed in order to render it suitable for publication:

- Whilst it is specified in the Discussion section that staining performed post MALDI-MSI analysis can be distorted by the impacts of laser irradiation (depending on the matrix properties and the laser fluence required), however, it has been reported that staining prior to lipid MSI may reduce sensitivity (doi: 10.3389/fchem.2024.1334209). Do the authors have any data from unstained counterparts in order to evaluate the impact that the immunostaining and various preparative washes might have on the sensitivity and the lipid annotations that may be lost as a result? It would be nice to see this information added or at least touched upon somewhat.
- It is unclear as to when imaging was performed in positive-ion mode, negative-ion mode, or both (on consecutive sections or on smaller ROIs of each tissue section). This is not particularly well described in the “Transmission-MALDI-2 MSI” section of the Materials and Methods and should be described more clearly (even if it is later inferred when describing the criteria for lipid annotation). Moreover, based upon Table S2, it would appear that imaging was only performed in positive-ion mode when mapping the 4T1-tumour mouse model. What was the rationale for excluding negative-ion mode imaging of this model?
- Within the main body of the manuscript, many of the lipid species presented in the figures relate to those generated in positive-ion mode. Is this merely a consequence of the MALDI matrices utilised in this study, and their associated properties, or as a result of the ionisation process being hindered by the previous immunostaining when working in this modality?
- It is referenced that the ability to detect phosphatidylethanolamine (PE) and phosphatidylserine (PS) lipids in formalin-fixed tissue is hindered given that their head groups are implicated within the cross-linking process and is supported by a citation from Vos et al. (doi.org/10.1016/j.ijms.2019.116212). However, in that work, tissue was fixed for 24 hours (in line with histopathological guidelines for tissue of that size) whilst here tissue fixation was seemingly performed for 5 minutes. Therefore, I would not be certain that such a short fixation time would notably hinder access to the aforementioned lipid species.

Minor comments:

- In line 28 of the Introduction, the term “genetic code” is used incorrectly. “Genetic code” specifically refers to the set of rules that translate the sequence of nucleotides in DNA into the sequence of amino acids in proteins. Therefore, all cells share the same genetic code. The correct terms to describe inherited genetic information passed down from ancestors are “genetic heritage” or “genetic makeup”.
- There are some minor typing and grammatical errors that could be resolved with further proofreading of the entire manuscript (ie. “is setup partly” should read “is set up partly”, “a few pixel” should read “a few pixels”). This is only a very

minor comment, however, that does not hinder the readability of the well-written manuscript.

(Remarks on code availability)

Reviewer #2

(Remarks to the Author)

(Remarks on code availability)

I co-reviewed this manuscript with one of the reviewers who provided the listed reports. This is part of the Nature Communications initiative to facilitate training in peer review and to provide appropriate recognition for Early Career Researchers who co-review manuscripts

Reviewer #3

(Remarks to the Author)

The work is well performed and technically excellent. The work should be published. The issue is that, as the authors state, the biology is not the driver of the effort but this report is driven by new technology. However, it is not clear what is new and what is similar to prior systems / reports.

After all, the combination of MSI and optical microscopy has been done for a long while. This includes both two separate instruments and systems that integrate optical microscopy and MSI. It has been commercialized, has been demonstrated with MSI and IR, Raman, and other optical systems. The integration of small molecule stains, many flavors of optical microscopy followed by MSI has been the topic of multiple prior publications. The use of optical microscopy to define cells (their cell segmentation masks) is not new.

Thus, while this system represents an improvement, especially in terms of the MSI, it is not clear what is novel. The authors state that the achieved combination of lipid profiling and morphological features and protein expression on the single-cell level constitutes a powerful new method for cell biology. Yes, the MSI is better than prior reports but that advance has been covered in prior publications. Advances by these authors and others have driven MSI to the 1 – 5 micron range to enable single cell MSI! Of course, optical microscopy has been achieving this resolution for many decades and its integration with MSI is well known. Said differently, it is not clear what the hyphenation really advances here compared to other recently previously published efforts.

(Remarks on code availability)

Version 1:

Reviewer comments:

Reviewer #1

(Remarks to the Author)

The Reviewers thank the Authors for their well thought out replies to our comments and the modifications made to the manuscript. On the whole, the manuscript can be accepted, however there are one or two minor details that could be ironed out prior to publication:

1. On line 473, there is a typo; “und” should read “and”.

2. Regarding Table S1, whilst the annotations, in terms of qualitative numbers appear to be similar, this data could be strengthened by showing a selection of ion images that also indicate that there is no significant change in ion abundance as the result of the statement. Moreover, after excluding possible matrix clusters, it would also be helpful to have an understanding of the number of peaks present in both conditions (ie. Also considering those unable to be annotated, given that these may also represent some of the more lower abundant lipid features present within the dataset and may be of greater importance in more clinically orientated studies)

Once again, compliments to the Authors for their work and this important contribution in the field.

(Remarks on code availability)

Reviewer #2

(Remarks to the Author)

(Remarks on code availability)

Reviewer #3

(Remarks to the Author)

Overall, the authors have done a nice job in responding to the reviewer comments. The manuscript reports exciting work and will interest a number of readers.

Two minor comments.

(1) While MALDI MSI works extremely well for lipids, no other molecular classes have been discussed, although the abstract / introduction both start by explaining that spatial biology is interested in gene and protein expression and MALDI MSI is used for lipid and metabolic profiles. This reviewer may have missed a showing of a metabolite in the results, but otherwise, the work is focused on lipids. While this is fine, perhaps the conclusions could expand on this in terms of future potential? After all, the ultimate spatial limitation of MSI is not the probe spot size but instrumental LOD; if the system LOD is 20 zmol, then a spot size that contains <20 zmol will show nothing. This is one reason that ubiquitous cellular lipids are often imaged at the smaller spot sizes compared to lower level metabolites. They cannot be observed.

This brings up a related point. What quality are the MS spectra? This reviewer is sure they are fine. However, it appears that neither the manuscript nor the SI include any mass spectra, perhaps outside of the heavily processed violin plots shown in one part of extended Figure 4. Showing several spectra to highlight both higher and lower SNR spectra used for the lipid annotations in the SI would be useful.

(Remarks on code availability)

Point-by-point response

Reviewer #1 (Remarks to the Author):

The authors Potthoff et al. present a manuscript entitled “Spatial biology using single-cell mass spectrometry imaging and integrated microscopy” in which they present a prototype transmission mode MALDI-2 ion source configuration that also allows for in-source bright field (BF) and fluorescence microscopy (FM) to be performed on a single sample, enabling coupled (sub-)cellular analysis using both the MSI and microscopic methods. This configuration represents a further step forward in terms of the instrumental developments presented by the research team in question and, in particular, responds to a rather evident need in the field related to challenges faced when performing multi-modal imaging and the precise co-registration needed in order to arrive at the single and, in particular, sub-cellular levels.

On the whole, the manuscript is written in a thorough manner and appears to be technically sound. Moreover, it seeks to address a key issue that hinders the further progress of this technology within the context of spatial biology, especially at the single-cell level, and therefore represents noteworthy research. Notwithstanding, there are some minor concerns that should be addressed in order to render it suitable for publication:

Response: We thank the reviewer for these positive comments. We are glad that this reviewer shares our assessment of current key challenges in MSI of single cells and approves of our approach to overcome some of them.

Comment: Whilst it is specified in the Discussion section that staining performed post MALDI-MSI analysis can be distorted by the impacts of laser irradiation (depending on the matrix properties and the laser fluence required), however, it has been reported that staining prior to lipid MSI may reduce sensitivity (doi: 10.3389/fchem.2024.1334209). Do the authors have any data from unstained counterparts in order to evaluate the impact that the immunostaining and various preparative washes might have on the sensitivity and the lipid annotations that may be lost as a result? It would be nice to see this information added or at least touched upon somewhat.

Response: We agree with the reviewer that pre-MALDI staining could potentially decrease spectral quality or depth of information. In fact, optimizing our workflow was centered on omitting or strongly reducing these effects. This resulted in considerable deviations from standard staining protocols, especially with regard to the use of solvent and detergents.

To strengthen this point, we have added additional data to the supporting information and a description to the main text that compares the number of lipid annotations between stained and naïve tissue sections of the same brain sample measured under comparable t-MALDI-MSI conditions (table S1). As observed previously, mass spectra in the untreated sample are dominated by alkali cation adducts in the positive ion mode, while stained and washed samples produce predominantly protonated lipid ion species. While this reduces the overall number of peaks, the number of lipid annotations and with it the depth of information remains comparable. Of note, similar effects can be observed in the above mentioned work of Shafer and Neumann based on figures presented there. Somewhat unfortunate, however, it seems that alkali metal cations were not considered in lipid annotation in that publication.

Comment: It is unclear as to when imaging was performed in positive-ion mode, negative-ion mode, or both (on consecutive sections or on smaller ROIs of each tissue section). This is not particularly well described in the “Transmission-MALDI-2 MSI” section of the Materials and Methods and should

be described more clearly (even if it is later inferred when describing the criteria for lipid annotation). Moreover, based upon Table S2, it would appear that imaging was only performed in positive-ion mode when mapping the 4T1-tumour mouse model. What was the rationale for excluding negative-ion mode imaging of this model? Within the main body of the manuscript, many of the lipid species presented in the figures relate to those generated in positive-ion mode. Is this merely a consequence of the MALDI matrices utilised in this study, and their associated properties, or as a result of the ionisation process being hindered by the previous immunostaining when working in this modality?

Response: We thank the reviewer for this heads-up. We have clarified the ion mode used for all experiments throughout the manuscript, including figure captions and the methods.

While the workflow generally works in both polarities, the combination with CHCA matrix and MALDI-2 produced more information in the positive ion mode. CHCA was chosen for its superior performance in t-MALDI concerning the size of ablation craters and overall signal intensities. A direct comparison with the negative ion mode demonstrated on the brain sample produced only a small number of lipid species, such as phosphatidylinositols, solely detected in negative ion mode (table S2). Because only one t-MALDI measurement can be performed on a single slide, we therefore chose positive ion mode for all other experiments. This is now clarified in the main text.

Comment: It is referenced that the ability to detect phosphatidylethanolamine (PE) and phosphatidylserine (PS) lipids in formalin-fixed tissue is hindered given that their head groups are implicated within the cross-linking process and is supported by a citation from Vos et al. (doi.org/10.1016/j.ijms.2019.116212). However, in that work, tissue was fixed for 24 hours (in line with histopathological guidelines for tissue of that size) whilst here tissue fixation was seemingly performed for 5 minutes. Therefore, I would not be certain that such a short fixation time would notably hinder access to the aforementioned lipid species.

Response: We agree with the reviewer that the short fixation in our protocol does not lead to signal reduction for PS and PE. Because it has been shown in the past, however, that longer chemical fixation can indeed lead to loss of lipid ion signal due to crosslinking, we intended to pre-emptively demonstrate that this is not the case for our protocol. To make this point clearer, we have amended the somewhat misleading description.

Comment: In line 28 of the Introduction, the term “genetic code” is used incorrectly. “Genetic code” specifically refers to the set of rules that translate the sequence of nucleotides in DNA into the sequence of amino acids in proteins. Therefore, all cells share the same genetic code. The correct terms to describe inherited genetic information passed down from ancestors are “genetic heritage” or “genetic makeup”.

Response: Fixed

Comment: There are some minor typing and grammatical errors that could be resolved with further proofreading of the entire manuscript (ie. “is setup partly” should read “is set up partly”, “a few pixel” should read “a few pixels”). This is only a very minor comment, however, that does not hinder the readability of the well-written manuscript.

Response: We thank the reviewer for these observations. The manuscript was carefully proofread and corrected accordingly.

Reviewer #2 (Remarks to the Author):

Reviewer #2 (Remarks on code availability):

Response: We thank reviewer 2 for their time and efforts.

Reviewer #3 (Remarks to the Author):

The work is well performed and technically excellent. The work should be published. The issue is that, as the authors state, the biology is not the driver of the effort but this report is driven by new technology. However, it is not clear what is new and what is similar to prior systems / reports. After all, the combination of MSI and optical microscopy has been done for a long while. This includes both two separate instruments and systems that integrate optical microscopy and MSI. It has been commercialized, has been demonstrated with MSI and IR, Raman, and other optical systems.

Response: We thank the reviewer for the overall positive assessment of our work and also for the comments about the novelty of our method. The reviewer is right in the statements that MALDI and microscopy have been combined for quite a long time in many different ways and in a large number of publications. It is, however, also reported frequently, that the lack of a highly precise co-registration at the cellular level is a major roadblock in the generation and analysis of mass spectral information of individual single cells, especially from tissue. For this task, the main difficulties lay in the unambiguous assignment of MSI pixels to a specific cell. While for the analysis of cultured and sub-confluent cells this problem is somewhat reduced, the analysis of dense tissue requires a small pixel size combined with accurate definition and co-registration of cell masks. This is what we present here for the first time.

In conventional setups, co-registration is often conducted based on fiducial markers across modalities. As described by Ščupáková et al. (<https://doi.org/10.1002/ange.202007315>) and others, accurate co-registration based on fiducial markers on a cellular level requires manual fine-tuning to achieve co-registration with an accuracy in the range of 5 μm .

Other approaches acquire microscopic measurements before and after MALDI-MSI and utilize laser ablation marks as fiducial markers. This approach however, requires undersampling conditions, where clear ablation marks are visible after analysis. This is usually not the case for high resolution measurements such as t-MALDI analysis at 1 μm pixel size. In addition, microscopic analysis after MALDI analysis requires washing of the matrix that may distort the tissue section and that may lead to other detrimental effects.

To circumvent the need for fiducial markers, microscopic analysis can be implemented into the mass spectrometer. Shimadzu (iMScope) commercialized this. In this setup, however, microscope and MSI ion source are spatially separated inside the instrument and the comparably long sample transfer between both modalities leads to a positioning error of approx. 5 μm . In addition, it is limited to bright field analysis that can be challenging for samples covered with MALDI matrix. In transmission mode MALDI setups, similar to the one used in our work, the optical setup has been used to observe the sample. In those cases it only provided a very limited field of view and has not been utilized for slide scanning microscopy. Again only bright field has been used.

In conclusion, the key challenges for the combination of fluorescence microscopy and high resolving power MALDI-MSI on the cellular level lays in a robust, precise and high fidelity co-registration across large regions of interest remains unresolved.

We believe that our approach is the first to overcome this challenge convincingly. This includes the specific analysis of sub-cellular structures as well as tens of thousands of cells from tissue.

Comment: The integration of small molecule stains, many flavors of optical microscopy followed by MSI has been the topic of multiple prior publications.

Response: We thank the reviewer for this comment. It has initiated a thorough reappraisal of the pertinent literature regarding small molecule stains conducted prior to MALDI. With one exception, however, the described instances of this method are limited to the analysis of cell-culture rather than tissue samples. The exception uses a concept dubbed DALDI where the dye is also used as a matrix and is rather unrelated to the work at hand (DOI: 10.1089/omi.2013.0175). The reference was therefore omitted from the manuscript to keep the focus. Other flavors of optical microscopy conducted prior to MALDI such as bright field, phase contrast or auto fluorescence of unstained tissue as well as methods such as IR or Raman microscopy have been added to the manuscript. In one instance, the use of small molecule as well as immunofluorescence prior to MALDI has been described but deemed unsuccessful by the authors of the study in question (doi: 10.3389/fchem.2024.1334209). This reference was also added.

Much more commonly, microscopy of many flavors and stains is conducted after MALDI analysis. Additional information about the benefits and drawbacks of this technique was also added to the manuscript.

Comment: The use of optical microscopy to define cells (their cell segmentation masks) is not new.

Response: We fully agree that cell segmentation based on optical microscopy is not new. In fact, we take full advantage of the impressive progress made in this field in recent years and the tools available for cell segmentation are vital in enabling us to generate large numbers cell masks. The novelty of our approach lies in the accurate transfer of these cell segmentation masks to MSI images of tissue with sufficient accuracy in order to generate unambiguous single cell spectra in an automated fashion. This novel approach is an advancement of similar approaches recently introduced for the analysis of cell culture under the name FISCAS by our group.

This is clarified in the text.

Comment: Thus, while this system represents an improvement, especially in terms of the MSI, it is not clear what is novel. The authors state that the achieved combination of lipid profiling and

morphological features and protein expression on the single-cell level constitutes a powerful new method for cell biology.

Response: We politely disagree with the reviewer that the combined mass spectrometric analysis and morphometric analysis including protein expression at the level of single cells has already been covered for a statistically meaningful number of cells. Prior attempts either have been limited by ambiguous annotation of pixels to single cell mass spectra caused by large MALDI pixels or inaccurate co-registration, laborious manual annotation of cells, or did not include IF based identification of specific cell types.

Comment: Yes, the MSI is better than prior reports but that advance has been covered in prior publications. Advances by these authors and others have driven MSI to the 1 – 5 micron range to enable single cell MSI! Of course, optical microscopy has been achieving this resolution for many decades and its integration with MSI is well known. Said differently, it is not clear what the hyphenation really advances here compared to other recently previously published efforts.

Response: Again, we politely disagree with the reviewer. We are convinced that only the effective combination of a number of factors is the key to success for the presented technique. These factors include pre-MALDI FM staining, in-source FM, 1 μm pixel size t-MALDI-2 analysis, and the respective software tools. While some, but not all, of these factors have been presented previously, it is only through their effective hyphenation and tight interplay that enables this type of (sub-) cellular analysis. Importantly, we would like to emphasize that combining even a few of these elements in a single experimental workflow is far from trivial and poses significant technical and methodological challenges. While it might be possible to generate single cell spectra based solely on 5 μm pixel size MSI from sub-confluent grown cell culture, single cells cannot be discerned or segmented in MSI images of tissue even at 1 μm pixel size. Consequently, it needs the hyphenation of 1 μm pixel size MSI data with microscopy and dedicated staining to allow for the generation and accurate transfer of cell masks to the MSI data. This hyphenation or co-registration, in turn, needs a precision of 1-2 μm to reduce ambiguity provided through the in-source microscopy. We are convinced that this interdependent hyphenation of many different factors along with the technical advances presented in this manuscript for the first time (e.g. in-source slide scanning fluorescence microscopy) constitutes a significant scientific work in itself that goes far beyond merely establishing a mere sequence of existing technologies as insinuated by the reviewer.

To emphasize the importance and novelty of this hyphenation, we have amended the manuscript. In line with the guides for authors of this journal, however, we have largely omitted the use of phrases such as new or novel.

We thank all reviewers for their valuable comments and their time and efforts

Point-by-Point reply to reviewer's comments:

Reviewer #1 (Remarks to the Author):

1. On line 473, there is a typo; "und" should read "and".

Reply: Corrected

2. Regarding Table S1, whilst the annotations, in terms of qualitative numbers appear to be similar, this data could be strengthened by showing a selection of ion images that also indicate that there is no significant change in ion abundance as the result of the statement. Moreover, after excluding possible matrix clusters, it would also be helpful to have an understanding of the number of peaks present in both conditions (ie. Also considering those unable to be annotated, given that these may also represent some of the more lower abundant lipid features present within the dataset and may be of greater importance in more clinically orientated studies)

Reply: We thank the reviewer for this suggestion and have added representative mass spectra to provide an unfiltered view of MSI data. As described in the main text and indicated in Supplementary Table 2, washing and staining of the tissue sections leads to a depletion of alkali metal salts and a shift in the detected ion species from primarily $[M+Na]^+/[M+K]^+$ to $[M+H]^+$. For this reason, we believe that the number of peaks is not an ideal measure to describe the analytical depth and have not included it into the manuscript.

Reviewer #3 (Remarks to the Author):

Two minor comments.

(1) While MALDI MSI works extremely well for lipids, no other molecular classes have been discussed, although the abstract / introduction both start by explaining that spatial biology is interested in gene and protein expression and MALDI MSI is used for lipid and metabolic profiles. This reviewer may have missed a showing of a metabolite in the results, but otherwise, the work is focused on lipids. While this is fine, perhaps the conclusions could expand on this in terms of future potential? After all, the ultimate spatial limitation of MSI is not the probe spot size but instrumental LOD; if the system LOD is 20 zmol, then a spot size that contains <20 zmol will show nothing. This is one reason that ubiquitous cellular lipids are often imaged at the smaller spot sizes compared to lower level metabolites. They cannot be observed.

Reply: While molecules such as DG, LPC, and LPG as well as ACAR are products of the lipid metabolisms and should be considered metabolites, we agree with the reviewer that the presented work is mostly centered on lipids and other unipolar analyte classes and mostly excludes more polar metabolite species. To comply with this limitation, we have amended the discussion to point.

"The seamless integration of morphometric data with (sub-)cellular lipid and **possibly metabolite** analysis significantly expands the informational space available for each individual cell inside the tissue."

"Embedded within a full suite of spatial biology methods, in future applications, it will allow for the combination of IF-based information that may describe the regulation of specific genes and the

expression of individual proteins with the expression of lipids and **possibly also metabolites** in the same cell and its direct surrounding”

In addition, we have added future directions and an outlook to the discussion that includes alternative strategies to sample preparation and post ionization to increase sensitivity for metabolite analysis.

“Future developments in t-MALDI-MSI will need to focus on further increasing the sensitivity and analytical depth of the method. E. g., innovative approaches to sample preparation combined with plasma-based ionization approaches, as described by Young et al. in the same issue of this journal, can help to increase sensitivity and assist in the analysis of additional classes of metabolites.”

This brings up a related point. What quality are the MS spectra? This reviewer is sure they are fine. However, it appears that neither the manuscript nor the SI include any mass spectra, perhaps outside of the heavily processed violin plots shown in one part of extended Figure 4. Showing several spectra to highlight both higher and lower SNR spectra used for the lipid annotations in the SI would be useful.

Reply: We thank the reviewer for this last minute reminder that MS based publications should indeed contain some mass spectra. Accordingly, we have added representative mass spectra with a number of exemplary annotations to Supplementary figure 2.